# AP-2α and AP-2β cooperatively function in the craniofacial surface ectoderm to regulate chromatin and gene expression dynamics during facial development

Eric Van Otterloo[1,2,3,4]*, Isaac Milanda[4], Hamish Pike[4], Jamie A Thompson[1,3], Hong Li[4], Kenneth L Jones[5†], Trevor Williams[4,6,7]*

[1]Iowa Institute for Oral Health Research, College of Dentistry & Dental Clinics, University of Iowa, Iowa City, United States; [2]Department of Periodontics, College of Dentistry & Dental Clinics, University of Iowa, Iowa City, United States; [3]Department of Anatomy and Cell Biology, Carver College of Medicine, University of Iowa, Iowa City, United States; [4]Department of Craniofacial Biology, University of Colorado Anschutz Medical Campus, Aurora, United States; [5]Department of Pediatrics, Section of Hematology, Oncology, and Bone Marrow Transplant, University of Colorado School of Medicine, University of Colorado Anschutz Medical Campus, Aurora, United States; [6]Department of Cell and Developmental Biology, University of Colorado Anschutz Medical Campus, Aurora, United States; [7]Department of Pediatrics, University of Colorado Anschutz Medical Campus, Children's Hospital Colorado, Aurora, United States

*For correspondence:
eric-vanotterloo@uiowa.edu
(EVO);
trevor.williams@cuanschutz.edu
(TW)

Present address: †Department of Cell Biology, University of Oklahoma Health Sciences Center, Oklahoma City, United States

**Abstract** The facial surface ectoderm is essential for normal development of the underlying cranial neural crest cell populations, providing signals that direct appropriate growth, patterning, and morphogenesis. Despite the importance of the ectoderm as a signaling center, the molecular cues and genetic programs implemented within this tissue are understudied. Here, we show that removal of two members of the AP-2 transcription factor family, AP-2α and AP-2ß, within the early embryonic ectoderm of the mouse leads to major alterations in the craniofacial complex. Significantly, there are clefts in both the upper face and mandible, accompanied by fusion of the upper and lower jaws in the hinge region. Comparison of ATAC-seq and RNA-seq analyses between controls and mutants revealed significant changes in chromatin accessibility and gene expression centered on multiple AP-2 binding motifs associated with enhancer elements within these ectodermal lineages. In particular, loss of these AP-2 proteins affects both skin differentiation as well as multiple signaling pathways, most notably the WNT pathway. We also determined that the mutant clefting phenotypes that correlated with reduced WNT signaling could be rescued by *Wnt1* ligand overexpression in the ectoderm. Collectively, these findings highlight a conserved ancestral function for AP-2 transcription factors in ectodermal development and signaling, and provide a framework from which to understand the gene regulatory network operating within this tissue that directs vertebrate craniofacial development.

## Editor's evaluation

The TFAP2 transcription factor family is a well known regulator of craniofacial development and evolution. In this study the authors have undertaken a comprehensive analysis of TFAP2a and TFAP2b function in the facial ectoderm. Through genomic analyses, the authors identified key

TFAP2 dependent genomic elements in the facial ectoderm, their predicted transcription factor binding profiles. WNT signaling is downregulated in TFAP2mutants and can be rescued by overexpressing Wnt1 in the facial ectoderm. Overall, this study provides new insights into the role of AP2 genes in facial ectoderm signaling and development during craniofacial morphogenesis.

## Introduction

The development of the vertebrate face during embryogenesis requires the integration of gene regulatory programs and signaling interactions across different tissue layers to regulate normal growth and morphogenesis (*Chai and Maxson, 2006*; *Dixon et al., 2011*). The bulk of the face is derived from neural crest cells (NCCs), which migrate into the nascent mandibular, maxillary, and frontonasal facial prominences. Recent studies have indicated the cranial NCCs (CNCCs), residing within distinct facial prominences, are molecularly similar, genetically poised, and awaiting additional signaling information for their continued development (*Minoux et al., 2017*; *Minoux and Rijli, 2010*). These critical signals are provided by surrounding and adjacent tissues, especially the forebrain, endoderm, and ectoderm (*Le Douarin et al., 2004*). With respect to the ectoderm, studies in chick have indicated the presence of a frontonasal ectodermal zone, defined by the juxtaposition of *Fgf8* and *Shh* expressing domains, that can direct facial outgrowth and patterning (*Hu and Marcucio, 2009*; *Hu et al., 2003*). The ectoderm is also a critical source of Wnt signaling that is required for continued facial outgrowth and patterning, exemplified by the lack of almost all craniofacial structures arising when *Wls* (*Gpr177*) is removed from the facial ectoderm (*Goodnough et al., 2014*; *Reynolds et al., 2019*). Further evidence for an essential role of the ectoderm in craniofacial development comes from genetic analysis of pathology associated with human syndromic orofacial clefting. Specifically, mutations in *IRF6* (*Kondo et al., 2002*) and *GRHL3* (*Peyrard-Janvid et al., 2014*) are associated with van der Woude Syndrome, while *TRP63* mutations result in ectodermal dysplasias with associated facial clefting (*Celli et al., 1999*). Notably, all three of these human genes encode transcription factors which exhibit much stronger expression in the facial ectoderm than in the underlying neural crest (*Hooper et al., 2020*; *Leach et al., 2017*). Studies of mouse facial dysmorphology have also shown the importance of additional genes with biased expression in the ectoderm—including *Sfn*, *Jag2*, *Wnt9b*, and *Esrp1*—that regulate differentiation, signaling, and splicing (*Bebee et al., 2015*; *Jiang et al., 1998*; *Jin et al., 2012*; *Lee et al., 2013*; *Lee et al., 2020*; *Richardson et al., 2006*). Indeed, the interplay between surface ectoderm and underlying NCCs provides a molecular platform for the craniofacial diversity apparent within the vertebrate clade, but also serves as a system which is frequently disrupted to cause human craniofacial birth defects. Therefore, identifying the regulatory mechanisms and factors involved in coordinating NCC:ectoderm interactions is a prerequisite for uncovering the molecular nodes susceptible to perturbation.

The AP-2 transcription factor family represents an intriguing group of regulatory molecules with strong links to ectodermal development (*Eckert et al., 2005*). Indeed, previous analyses have indicated that AP-2 genes may be an ancestral transcriptional regulator of ectoderm development in chordates predating the development of the neural crest in the cephalochordate Amphioxus and the ascidian *Ciona* (*Imai et al., 2017*; *Meulemans and Bronner-Fraser, 2002*; *Meulemans and Bronner-Fraser, 2004*). Subsequently, it has been postulated that this gene family has been co-opted into the regulatory network required for neural crest development in the vertebrates, where it may serve as one of the master regulators of this lineage (*Meulemans and Bronner-Fraser, 2002*; *Meulemans and Bronner-Fraser, 2004*; *Van Otterloo et al., 2012*). Therefore, in vertebrates, AP-2 family expression is often observed in both the non-neural ectoderm as well as the neural crest. Amphioxus possesses a single AP-2 gene, but in mammals such as mouse and human there are five family members, *Tfap2a-e* encoding the proteins AP-2α-ε, respectively (*Eckert et al., 2005*; *Meulemans and Bronner-Fraser, 2002*). All mammalian AP-2 proteins have very similar DNA sequence preferences and bind as dimers to a consensus motif GCCNNNGGC, except for AP-2δ which is the least conserved family member (*Badis et al., 2009*; *Williams and Tjian, 1991*; *Zhao et al., 2001*). Amongst these five genes, *Tfap2a* and *Tfap2b* show the highest levels of expression in the developing mouse embryonic facial tissues with lower levels of *Tfap2c* and essentially undetectable transcripts from *Tfap2d* and *Tfap2e* (*Hooper et al., 2020*; *Van Otterloo et al., 2018*). Importantly, mutations in human *TFAP2A* and *TFAP2B*, are also linked to the human conditions Branchio-Oculo-Facial Syndrome (*Milunsky et al., 2008*) and

Char Syndrome (*Satoda et al., 2000*) respectively, conditions which both have a craniofacial component. *TFAP2A* has also been linked to non-syndromic orofacial clefting (MIM 119530) (*Davies et al., 1995*; *Davies et al., 2004*).

Previous single mouse knockout studies have indicated that the loss of *Tfap2a* has the most significant effect on craniofacial development with most of the upper face absent as well as split mandible and tongue (*Schorle et al., 1996*; *Zhang et al., 1996*). *Tfap2b* knockouts do not have gross morphological defects associated with craniofacial development (*Hong et al., 2008*; *Moser et al., 1997*; *Zhao et al., 2011*), nor do pertinent knockouts of any of the three other AP-2 genes (*Feng et al., 2009*; *Guttormsen et al., 2008*; *Hesse et al., 2011*). We have further investigated the tissue specific requirements for *Tfap2a* in face formation and determined that its loss in the neural crest resulted in cleft palate, but otherwise only minor defects in the development of the facial skeleton (*Brewer et al., 2004*). Next, we investigated whether the co-expression of *Tfap2b* might compensate for the loss of *Tfap2a* alone by deriving mice lacking both genes in NCCs. Although these NCC double knockout mice had more severe craniofacial defects, including a split upper face and mandible, the phenotype was still less severe than that observed with the complete loss of *Tfap2a* alone (*Van Otterloo et al., 2018*; *Zhang et al., 1996*). In contrast, targeting *Tfap2a* in the surface ectoderm in the region of the face associated with the lens placode causes a mild form of orofacial clefting (*Pontoriero et al., 2008*). These findings suggested that the ectoderm may be an additional major site of *Tfap2a* action during mouse facial development, and by analogy with the NCC studies, that the phenotype could be exacerbated by the additional loss of *Tfap2b*.

Therefore, here we have assessed how craniofacial development is affected upon simultaneous removal of *Tfap2a* and *Tfap2b* in the embryonic ectoderm using the Cre transgene, Crect, which is expressed from E8.5 onwards throughout this tissue layer. Our results show that the expression of these two AP-2 proteins in the ectoderm has a profound effect on the underlying NCC-derived craniofacial skeleton and strengthens the association between the AP-2 family and ectodermal development and function. Furthermore, we examined how the loss of these two AP-2 transcription factors impacted the ectodermal craniofacial gene regulatory network by studying changes in chromatin accessibility and gene expression between control and mutant mice. These studies reveal critical targets of AP-2 within the facial ectoderm, especially WNT pathway genes, and further indicate the necessity of appropriate ectodermal:mesenchymal communication for growth, morphogenesis, and patterning of the vertebrate face.

## Results
### Combined loss of *Tfap2a* and *Tfap2b* in the embryonic surface ectoderm causes major craniofacial defects

Previous studies have shown that *Tfap2a* and *Tfap2b* have overlapping functions within the neural crest in regulating facial development (*Van Otterloo et al., 2018*) raising the possibility that these transcription factors might also act together in the overlying surface ectoderm to regulate this aspect of embryogenesis. Therefore, we documented expression of the five family members in the ectoderm of the facial prominences based on analysis of previous RNAseq datasets spanning E10.5 and E12.5 (*Hooper et al., 2020*). *Tfap2a* and *Tfap2b* were the most highly expressed in the ectoderm, with lower levels of *Tfap2c*, and undetectable levels of *Tfap2d* and *Tfap2e* (*Figure 1A*). Further mining of single-cell RNA-seq data derived from facial prominences indicated that *Tfap2a* and *Tfap2b* expression also displayed significant overlap within cells of the surface ectoderm and periderm (*Figure 1B*). Since these two genes were the most highly expressed family members and were frequently expressed in the same cells, we next tested whether these two genes performed similar joint functions in the surface ectoderm in controlling growth and patterning as they do within the neural crest (*Van Otterloo et al., 2018*). Here the ectoderm expressed Cre recombinase transgene Crect (*Schock et al., 2017*) was used in concert with floxed versions of *Tfap2a* (*Brewer et al., 2004*) and *Tfap2b* (*Van Otterloo et al., 2018*) to remove these two transcription factors (TFs) from the early ectoderm. Using scanning electron microscopy, we found that at E11.5 both control and mutant embryos—hereafter designated ectoderm double knockout (EDKO)—had a similar overall facial organization, with distinct paired mandibular, maxillary, lateral and medial nasal processes (*Figure 1C–F*). However, there were also clear changes in the size and shape of these processes in the EDKO. The mandible was smaller with a

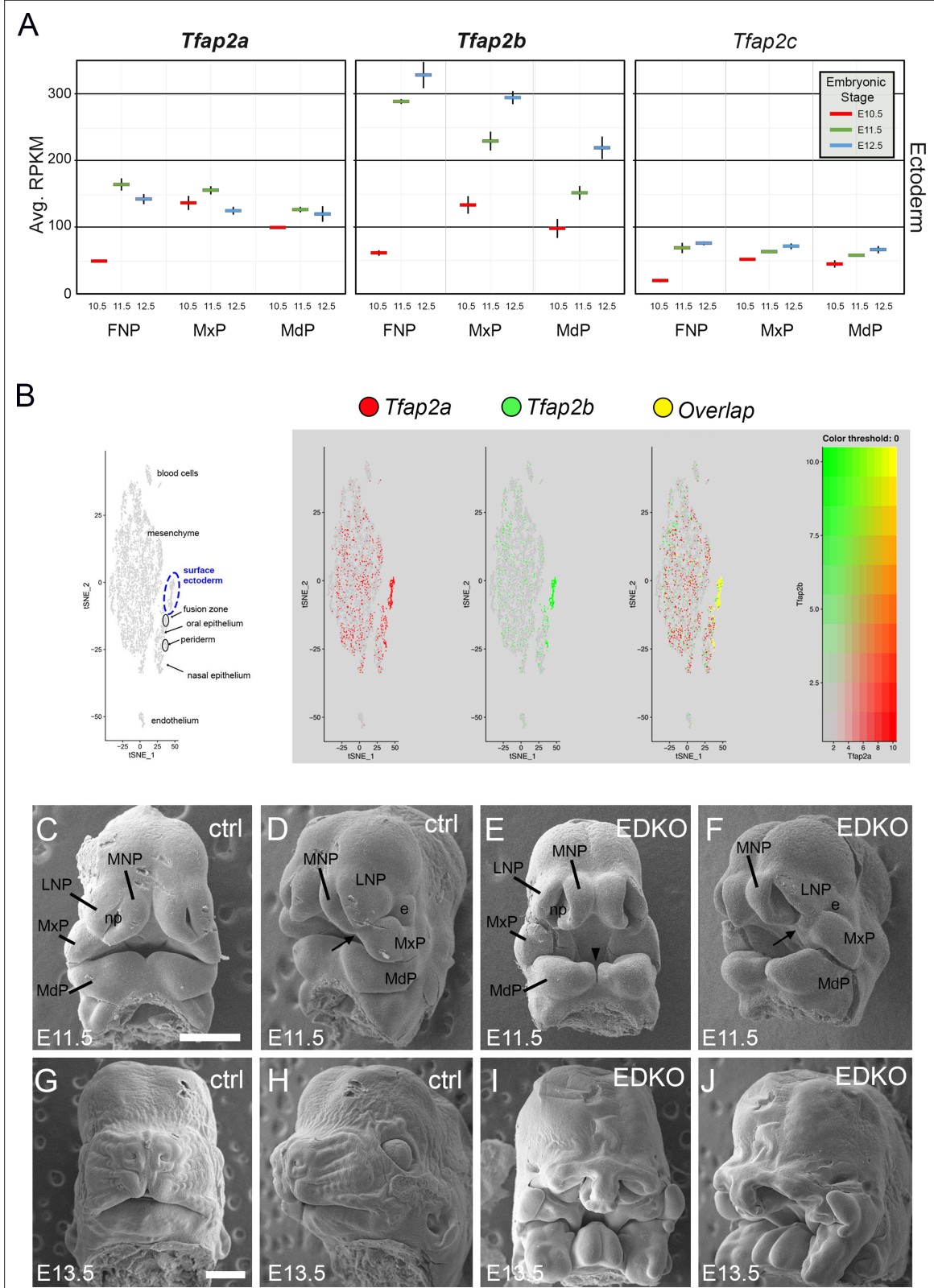

**Figure 1.** Expression and function of *Tfap2a* and *Tfap2b* in embryonic mouse facial ectoderm. (**A**) Chart depicting *Tfap2a*, *Tfap2b*, and *Tfap2c* expression in the three regions of the mouse ectoderm between E10.5 and E12.5 (data adapted from *Hooper et al., 2020*). The lines represent the standard deviation between three biological replicates. (**B**) Left panel shows tSNE plot of E11.5 single cell RNAseq data from the region surrounding the lambdoid junction with various cell populations labeled adapted from *Li et al., 2019a*. Feature plots for *Tfap2a*, *Tfap2b*, and the cells in which their

*Figure 1 continued on next page*

*Figure 1 continued*

expression overlaps are shown in the adjacent panels. (**C–J**) Scanning electron microscope images of E11.5 (**C–F**) or E13.5 (**G–J**) control (**C, D, G, H**) or EDKO (**E, F, I, J**) heads shown in frontal (**C, E, G, I**) and angled (**D, F, H, J**) view. Abbreviations: e, eye; FNP, combined nasal prominences; LNP, lateral nasal process; MdP, mandibular prominence; MNP, medial nasal process; MxP, maxillary prominence; np, nasal pit. Arrow shows position of lambdoid junction; arrowhead shows medial cleft between mandibular prominences in EDKO mutant. Ctrl embryos are *Tfap2a^flox/+*; *Tfap2b^flox/+* and EDKO embryos are Crect; *Tfap2a^flox/flox*; *Tfap2b^flox/flox*. N = 3 for each genotype. Scale bar = 500 µm.

The online version of this article includes the following figure supplement(s) for figure 1:

**Figure supplement 1.** Confirmation of Crect expression domains in the craniofacial ectoderm.

**Figure supplement 2.** IGV browser screenshot of RNA-seq tracks from E11.5 control (black) or EDKO (red) facial ectoderm.

more noticeable notch at the midline while in the upper face the maxilla and nasal processes had not come together to form a three-way lambdoid junction, and the nasal pit was more pronounced. By E13.5 these earlier morphological changes in the EDKOs were greatly exacerbated typified by a fully cleft mandible and a failure of the maxillary prominence (MxP), lateral nasal prominence (LNP), and medial nasal prominence (MNP) to undergo any productive fusion (*Figure 1G–J*). These observations indicate that the AP-2 TFs, particularly AP-2α and AP-2β, are critical components of a craniofacial ectodermal gene regulatory network (GRN). In the next section, we analyze this GRN in more detail, prior to describing additional analysis of the EDKO mouse model at later time points.

## ATAC-Seq of control and AP-2 mutant mouse craniofacial ectoderm identifies a core subset of unique nucleosome-free regions, many of which are AP-2 dependent

To investigate this GRN—and AP-2's potential role within it—we implemented ATAC-seq (*Buenrostro et al., 2013*; *Buenrostro et al., 2015*; *Corces et al., 2017*) on surface ectoderm pooled from the facial prominences of E11.5 control or EDKO embryos, processing two biological replicates of each (*Figure 2A*). We choose E11.5 for analysis since at this timepoint differences in craniofacial morphology between controls and mutants were becoming evident but were not yet severe (*Figure 1C–F*). To assess open chromatin associated with the craniofacial ectoderm GRN, we first focused our analysis on the control ectoderm datasets. From the combined control replicates, ~ 65 K (65,467) 'peaks' were identified above background (*Figure 2B*) representing open chromatin associated with diverse genomic *cis*-acting elements including promoters and enhancers. These elements were further parsed using ChIP-Seq data from E10.5 and E11.5 craniofacial surface ectoderm obtained using an antibody detecting the active promoter histone mark, H3K4me3. Specifically, the ATAC-seq peaks were classified into two distinct clusters, either high (N = 10,363) or little to no (N = 54,935) H3K4me3 enrichment (*Figure 2B*). Assessing the location of these peak classes relative to the transcriptional start site of genes clearly delineated them into either proximal promoter or more distal elements, respectively (*Figure 2C*). Motif enrichment analysis for the proximal promoter elements (*Andersson and Sandelin, 2020*) identified binding sites for Ronin, SP1, and ETS-domain TFs (*Figure 2D*, top panel, *Figure 2— source data 1*). Conversely, the top four significantly enriched motif families in distal elements were CTCF/BORIS, p53/63/73, TEAD, and AP-2 TFs (*Figure 2D*, bottom panel, *Figure 2—source data 2*). The most significant motif, CTCF/BORIS, is known to be found at insulator elements and is important in establishing topologically associated domains (*Dixon et al., 2012*; *Ong and Corces, 2014*). Notably, p53/63/73, TEAD, and AP-2 family members are highly enriched in open chromatin regions associated with early embryonic skin (*Fan et al., 2018*) and are known to be involved in skin development and often craniofacial morphogenesis (*Wang et al., 2006*; *Wang et al., 2008*; *Yuan et al., 2020*). Finally, pathway analysis of genes associated with either H3K4me3+ (*Figure 2—source data 3*) or H3K4me3- (*Figure 2—source data 4*) elements identified clear biological differences between these two subsets, with craniofacial and epithelial categories being prominent only in the latter.

We next reasoned that the H3K4me3- distal peaks likely represented regions of open chromatin that were found in multiple tissue-types as well as some that were ectoderm specific. Therefore, we utilized publicly available ATAC-seq datasets (*ENCODE Project Consortium, 2012*; *Davis et al., 2018*) from additional mouse embryonic tissues (liver, kidney, intestine, brain, etc.) and plotted relative peak intensities on top of our ~ 55 K distal peaks in the craniofacial surface ectoderm. K-means clustering of this overlap identified three distinct groups: 'tissue generic' (termed C1, N = 9244);

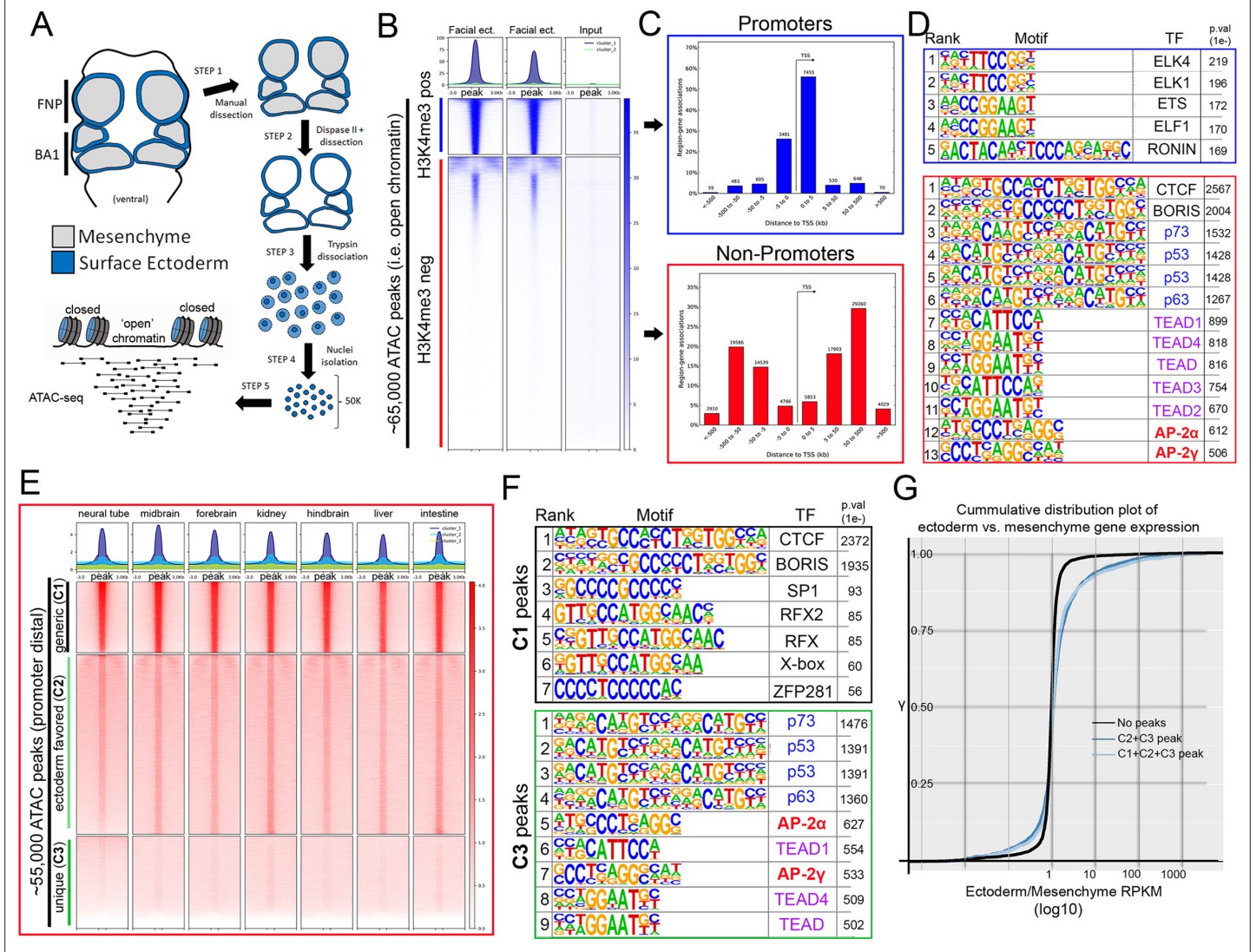

**Figure 2.** ATAC-seq of control E11.5 craniofacial surface ectoderm reveals nucleosome free regions. (**A**) A schematic outlining the general workflow of craniofacial surface ectoderm isolation and subsequent ATAC-seq to identify open chromatin regions. (**B**) Density plot of ~65,000 open chromatin regions identified in the control surface ectoderm (Y-axis), +/- 3 Kb (X-axis), overlaid with the H3K4me3 promoter mark from similar tissue at E10.5 (column 1), E11.5 (column 2), or non-enriched input control (column 3). (**C**) Distribution, relative to the transcriptional start site (TSS, arrow) of the elements subset in (**B**). (**D**) Transcription factor motif enrichment analysis of the 2 subset clusters identified in (**B**). (**E**) Density plot of ~55,000 non-promoter, open chromatin regions [bottom cluster in (**B**) replotted on Y-axis], +/- 3 Kb (X-axis) overlaid with ENCODE ATAC-seq datasets from various mouse embryonic tissues/organs. (**F**) Transcription factor motif enrichment analysis of 2 (C1 and C3) of the three subset clusters identified in (**E**) (C2 not shown). (**G**) A cumulative distribution plot of gene expression in craniofacial surface ectoderm versus mesenchyme. The groups of genes include those with no peaks (black line), those with C1, C2, and C3 peaks (light blue line), and those with C2 and C3 peaks only (dark blue line)—with 'peaks' being those defined by subclusters in (**E**).

The online version of this article includes the following source data for figure 2:

**Source data 1.** Summary of motif enrichment found within H3K4me3 + ATAC seq elements (i.e., *Figure 2D*, top).

**Source data 2.** Summary of motif enrichment found within H3K4me3- ATAC-seq elements (i.e., *Figure 2D*, bottom).

**Source data 3.** Summary of GREAT analysis of H3K4me3 + ATAC seq elements.

**Source data 4.** Summary of GREAT analysis of H3K4me3- ATAC-seq elements.

**Source data 5.** Summary of motif enrichment found within C1 ATAC-seq elements (i.e., *Figure 2E*, top).

**Source data 6.** Summary of GREAT analysis of C1 ATAC-seq elements (i.e., *Figure 2E*, top).

**Source data 7.** Summary of motif enrichment found within C3 ATAC-seq elements (i.e., *Figure 2E*, bottom).

**Source data 8.** Summary of GREAT analysis of C3 ATAC-seq elements (i.e., *Figure 2E*, bottom).

*Figure 2 continued on next page*

*Figure 2 continued*

**Source data 9.** Summary of motif enrichment found within C2 ATAC-seq elements (i.e., *Figure 2E*, middle).

**Source data 10.** Summary of GREAT analysis of C2 ATAC-seq elements (i.e., *Figure 2E*, middle).

**Source data 11.** A cumulative distribution plot graphing E11.5 craniofacial gene expression enrichment (ectoderm/mesenchyme, X-axis) relative to the total number of C2 and C3 ATAC-seq elements associated with that gene.

'ectoderm favored' (chromatin open in surface ectoderm, but also at low levels in other tissues, termed C2, N = 24,805); and 'ectoderm unique' (termed C3, N = 20,886) (*Figure 2E*). Motif analyses of these three subgroups showed that C1 was most highly enriched for the CTCF/BORIS motif (*Figure 2F*, *Figure 2—source data 5*) and genes nearby these elements had less relevant ectodermal/craniofacial associations (*Figure 2—source data 6*). Conversely, C3 elements contained the p53/p63/p73, AP-2, and TEAD motifs (*Figure 2—source data 7*), and nearby genes were highly enriched for networks associated with ectodermal and craniofacial development (*Figure 2F*, *Figure 2—source data 8*). In addition, the GRHL and PBX motifs—both key TF families in surface ectoderm gene networks (*Ferretti et al., 2011*; *Ting et al., 2005*)—were the next identified within the C3 element list at high significance. The C2 list contained a mix of both C3 and C1 motifs (*Figure 2—source data 9*) and gene network associations (*Figure 2—source data 10*).

Next, we employed the corresponding E11.5 gene expression profiles of the mouse craniofacial ectoderm and mesenchyme (*Hooper et al., 2020*) and correlated the relative expression between these two tissue layers with the list of E11.5 genomic elements and associated genes identified using ATAC-seq. Genes from the expression analysis were first binned into groups (*Supplementary file 1*) based upon whether they had: no associated peaks; peaks associated only with C1 (tissue generic), C2 (ectoderm favored), or C3 (ectoderm unique); or peaks in multiple categories (e.g. C1 + C2). We then used a cumulative distribution plot to assess the difference in distribution of 'ectoderm expression enrichment' between each group. This analysis identified that genes associated with both a C2 and C3 element showed a shift in distribution favoring ectoderm enrichment relative to genes with no associated element (p < 2.2e-16) (*Figure 2G*). In addition, if genes were also binned based on the sum of associated C2 and C3 elements, genes with 4 or greater elements, compared to those with less than 4, showed the most significant shift in distribution relative to genes with no elements (*Figure 2—source data 11*). Collectively, these analyses identified the position of key genomic elements in the mammalian craniofacial surface ectoderm, their predicted TF binding profiles, and correlation with ectoderm specific gene expression patterns and pathways. Moreover, these data suggested that AP-2 binding sites within promoter distal elements of ectodermally expressed genes may play an important role in the associated GRN required for facial development.

## Simultaneous loss of *Tfap2a* and *Tfap2b* within the surface ectoderm results in reduced chromatin accessibility at a subset of elements, including those associated with WNT ligands

To examine how loss of *Tfap2a* and *Tfap2b* impacted chromatin accessibility in the craniofacial ectoderm, we next analyzed the ATAC-seq data from the EDKO samples and compared the results to those obtained from controls. Combined analysis of the two EDKO samples yielded ~63,000 'peaks' with CTCF, P53/P63/P73, and TEAD again the top motifs identified (*Figure 3—source data 1*). In stark contrast to controls though, AP-2 consensus motifs were not detected, consistent with the loss of elements directly bound by AP-2 in EDKO mutants. Further, these data suggest that the limited expression of AP-2γ/*Tfap2c* in the ectoderm is not sufficient to compensate for the loss of AP-2α and AP-2β. Next, using the mutant dataset as 'background' to remove regions with similar chromatin accessibility from the control dataset, we identified genomic loci where accessibility was significantly higher in controls relative to in EDKO mutants. This differential analysis identified ~3.1 K genomic regions (N = 3103, ~ 5% of control elements) that were significantly decreased in accessibility upon loss of AP-2α/AP-2β (*Figure 3A*). AP-2 elements were the top two binding motifs in these 3.1 K peaks, consistent with AP-2 directly binding many of these elements (*Figure 3B*, *Figure 3—source data 2*). A more limited enrichment for p53/63/73, TEAD, and PBX motifs was also observed in these 3.1 K peaks, potentially indicating that AP-2 either facilitates access of these other TFs at certain sites or

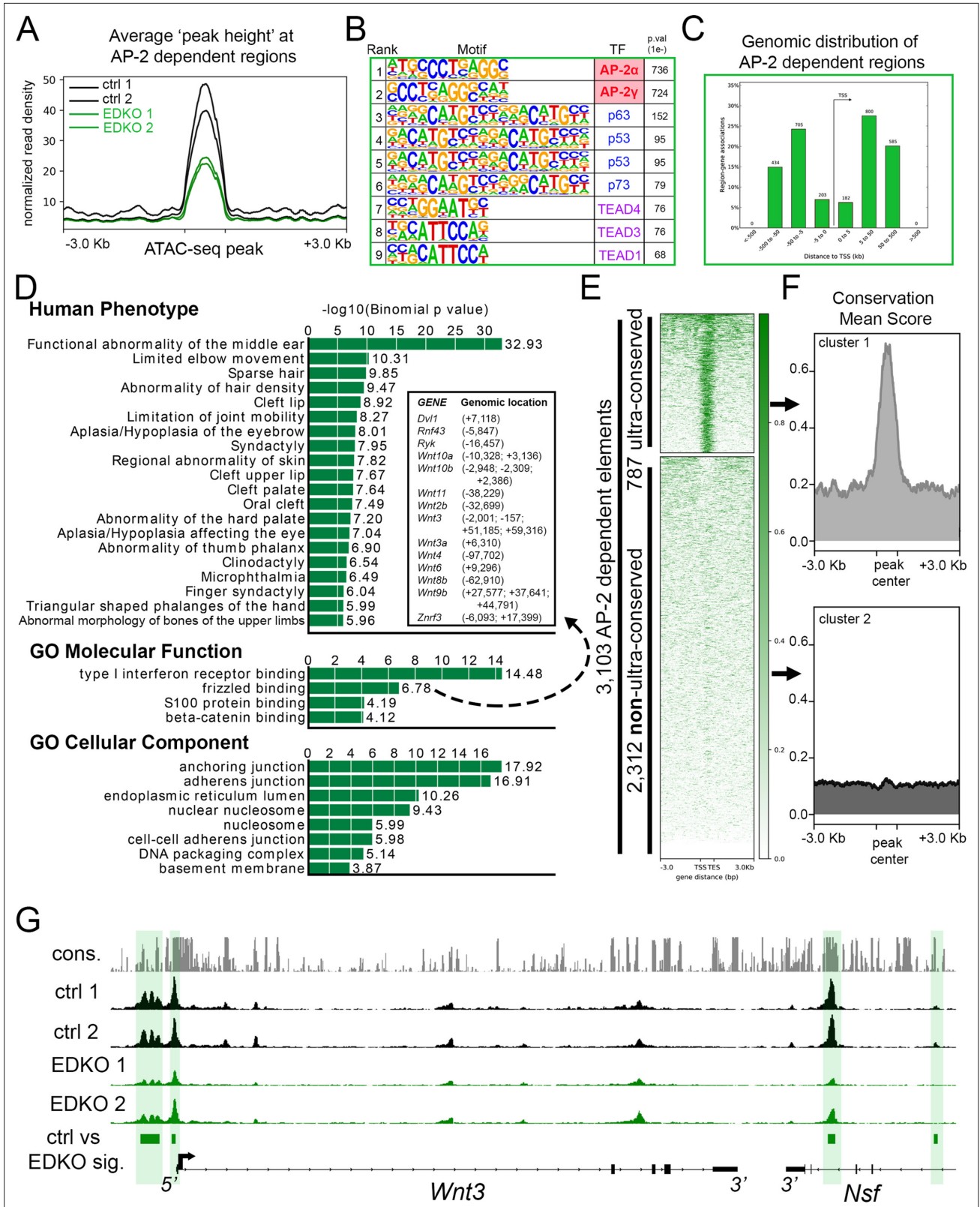

**Figure 3.** ATAC-seq analysis of EDKO mutants reveals AP-2 craniofacial surface ectoderm dependent nucleosome free regions. (**A**) Average normalized read density for control (black lines) and *Tfap2a/Tfap2b* ectoderm mutant (green lines) ATAC-seq datasets at AP-2-dependent nucleosome-free regions ( +/- 3.0 Kb). (**B**) Transcription factor motif enrichment analysis of AP-2-dependent nucleosome-free regions. (**C**) Distribution, relative to the transcriptional start site (TSS, arrow) of AP-2 dependent nucleosome-free regions. (**D**) GO/pathway enrichment analysis, using GREAT, of genes located near AP-2

*Figure 3 continued on next page*

*Figure 3 continued*

dependent nucleosome-free regions. Note, the inset highlights the genes associated with the GO Molecular Function annotation 'frizzled binding' and the genomic location (relative to the TSS) of the associated AP-2-dependent nucleosome-free region. (**E**) Density plot of ~3100 AP-2-dependent elements (Y-axis), +/- 3 Kb (X-axis) overlaid with conservation score (e.g. darker green = more conserved) identifies 'ultra-conserved' and 'non-ultra-conserved' subclusters. (**F**) Mean conservation score of elements identified in each subcluster in (**E**). (**G**) IGV browser view of tracks at the *Wnt3* locus. Tracks for conservation (grey, labeled *cons*.), control ATAC-seq replicates (black, labeled ctrl 1 and ctrl 2), AP-2 mutant ATAC-seq replicates (green, labeled EDKO1 and EDKO2), and coordinates of significantly altered elements between control and AP-2 mutant datasets (green bars, labeled ctrl vs EDKO sig.). The *Wnt3* transcription unit is schematized at the bottom, along with the 3' exons of the flanking *Nsf* gene, representing ~60 kb of genomic DNA.

The online version of this article includes the following source data for figure 3:

**Source data 1.** Summary of motif enrichment found within ATAC-seq elements remaining in EDKO mutant surface ectoderm.

**Source data 2.** Summary of motif enrichment found within ATAC-seq elements that are AP-2 -dependent (i.e., present in control, but gone in EDKO) in the craniofacial surface ectoderm.

**Source data 3.** Summary of GREAT analysis using ATAC-seq elements that are AP-2 dependent (i.e., present in control, but gone in EDKO) in the craniofacial surface ectoderm.

**Source data 4.** Summary of GREAT analysis using ATAC-seq elements that are AP-2 dependent (i.e., present in control, but gone in EDKO) in the craniofacial surface ectoderm and are 'ultra-conserved' (i.e., ***Figure 3E***, Top).

**Source data 5.** Summary of GREAT analysis using ATAC-seq elements that are AP-2 dependent (i.e., present in control, but gone in EDKO) in the craniofacial surface ectoderm and are 'non-ultra-conserved' (i.e., ***Figure 3E***, Bottom).

**Source data 6.** Summary of motif enrichment found within ATAC-seq elements that are gained upon loss of AP-2 in the craniofacial surface ectoderm (i.e., element not found in control, but present in EDKO).

simply reflecting the prevalence of these additional motifs in ectodermal control elements (***Figure 3B***, ***Figure 3—source data 2***).

Examination of this core subset of AP-2-dependent nucleosome free regions in the craniofacial ectoderm revealed that they are mostly promoter distal (~87%), consistent with enhancers (***Figure 3C***). Most genes (2,432) had only one assigned peak (***Supplementary file 2***), but many had two (654), three (232), four (108), or five (32) peaks. Notably, 45 genes had 6 or more assigned peaks, and ~120 peaks were assigned to only four gene pairs: *Rhou/Gas8*, *Ezh2/Pdia4*, *Atg7/Hrh1*, and *Asmt/Mid1*. However, these highly clustered assignments of 20–56 peaks per gene pair represent binding to direct repeat sequences, which skews functional annotations assigned by GREAT (***Figure 3D*** and ***Figure 3—source data 3***). Nevertheless, multiple genes and annotations associated with development of the skin and its appendages are still present (***Supplementary file 2*** and ***Figure 3—source data 3***). Thus, AP-2-dependent peaks had annotations including *anchoring junction* and *adherens junction* and were associated with genes encoding keratins, cadherins, and gap junction components (***Figure 3D***). Similarly, GO 'Molecular Function' annotations included both *frizzled binding* and *beta-catenin binding*, and multiple WNT pathway genes were also assigned to peaks (***Figure 3D***: *Wnt2b*, *Wnt3*, *Wnt3a*, *Wnt4*, *Wnt6*, *Wnt8b*, *Wnt9b*, *Wnt10a*, and *Wnt10b*)—some of which are known to be essential for proper craniofacial development (***Chiquet et al., 2008***; ***Menezes et al., 2010***; ***Reynolds et al., 2019***; ***Watanabe et al., 2006***).

Next, we further subdivided the AP-2-dependent elements based on their overall degree of conservation across vertebrate lineages (60-way phast-con score), creating two distinct clusters, 'ultra-conserved' (N = 787 elements) and less conserved (N = 2312 elements) (***Figure 3E and F***). Pathway analysis of genes associated with the ultra-conserved elements now revealed *frizzled binding* as the top 'Molecular Function'—in part, because of ultra-conserved elements near *Wnt3*, *Wnt9b*, *Wnt10a*, and *Wnt10b* (***Figure 3G*** and ***Figure 3—source data 4***). Interestingly, the only 'Human Phenotype' listed in the non-ultra-conserved group was '*cleft lip*', in part because of elements near the *Irf6* and *Grhl3* loci, but no WNT-related categories were identified in this list (***Figure 3—source data 5***). These findings suggest that distinct 'ancient' and 'derived' AP-2 networks exist in the craniofacial surface ectoderm. Finally, we utilized the control dataset as 'background' to look for enrichment in the EDKO dataset. This approach identified ~1.5 K regions that became more accessible upon loss of *Tfap2a* and *Tfap2b*, but motif analysis of these elements did not identify an enrichment of the AP-2-binding site, suggesting that direct AP-2 DNA binding is not responsible for blocking these sites in control ectoderm (***Figure 3—source data 6***).

In summary, our analysis of chromatin accessibility in AP-2 mutant craniofacial surface ectoderm suggests that: (1) a subset of distal nucleosome-free regions—presumed enhancers—is AP-2 dependent; (2) these elements are significantly enriched near genes regulating craniofacial and ectodermal development; (3) elements near WNT-related loci are disproportionally impacted upon loss of AP-2; and, (4) AP-2 regulation of chromatin dynamics near WNT-loci is likely a highly conserved function.

## Reduced chromatin accessibility at WNT-related genes correlates with reduced gene expression at E11.5 in EDKO surface ectoderm

Analysis of chromatin accessibility in EDKO mutants and controls indicated that loss of AP-2 in the ectoderm may impact expression of several genes in the WNT pathway. Therefore, at this juncture, we surveyed the distribution of multiple WNT pathway components in both the facial ectoderm and mesenchyme to ascertain how they correlated with expression of *Tfap2a* and *Tfap2b* using previously published bulk RNAseq (*Hooper et al., 2020*) or scRNAseq (*Li et al., 2019a*) datasets. This data mining confirmed that genes encoding Wnt ligands *Wnt3*, *Wnt4*, *Wnt9b* as well as the antagonist *Dkk4* showed biased expression in the surface ectoderm that overlapped on a cellular level with the two *Tfap2* genes (*Supplementary file 3* and *Figure 4—figure supplement 1*). Several other Wnt pathway genes, such as *Axin2* and *Sostdc1* showed notable overlap with the *Tfap2* genes in the surface ectoderm but were also present at significant levels in the underlying mesenchyme. The connection between AP-2 transcription factors and these Wnt pathway genes was further investigated using both real-time RT-PCR and RNA in situ hybridization to compare expression in E11.5 embryos between control and EDKO mutants. To extend the analysis, gene expression was also analyzed in embryos with additional *Tfap2a/Tfap2b^Crect* allelic combinations, specifically those lacking both copies of *Tfap2a*, but still containing one functional allele of *Tfap2b* (EAKO), and those with one functional allele of *Tfap2a*, but no *Tfap2b* (EBKO). In situ hybridization for *Wnt3* and *Wnt9b* in control embryos demonstrated strong expression in the facial ectoderm, typified by the signal observed at the margins of the MxP (*Figure 4A and E*). This staining was absent in the EDKO mutants (*Figure 4C and G*), and the EAKO mutants showed an intermediate level of staining (*Figure 4B and F*). RT-PCR analysis of E11.5 whole facial tissue confirmed these in situ findings for the ectodermally expressed ligands *Wnt3* and *Wnt9b*, as well as *Wnt10b* (*Figure 4D, H, I*). RT-PCR also revealed a graded reduction in expression from control, to EAKO, and finally EDKO mutants, for these three genes but no significant loss of expression in EBKO mutants, where an intact allele of *Tfap2a* was still present. Several WNT-signaling repressors—for example, *Axin2*, *Dkk4*, and *Sostdc1*—were also associated with elements showing reduced chromatin accessibility in facial ectoderm of EDKO mutants (*Figure 4M* and *Figure 4—figure supplements 2 and 3*). RT-PCR analysis of these 3 genes also showed reduced expression, especially between control and EDKO mutants (*Figure 4J–L*). Since *Axin2* has similar expression in ectoderm and mesenchyme (*Leach et al., 2017*), we next used RT-PCR to examine *Axin2* expression in the separated tissue layers of control and EDKO samples, in comparison to *Wnt3*, which exhibits mainly ectodermal expression (*Figure 4—figure supplement 4*). These studies showed that *Wnt3* down-regulation was confined to the ectoderm, whereas *Axin2* expression was reduced in both tissues, suggesting that AP-2 loss in the ectoderm may also be indirectly affecting the mesenchyme gene expression program. We further examined the impact of changes in epithelial:mesenchymal interactions caused by loss of *Tfap2a/Tfap2b* in the ectoderm by studying cell proliferation in the facial prominences of E11.5 control and EDKO embryos. As shown in *Figure 4—figure supplement 5*, α-phospho-Histone H3 (αPHH3) immuno-fluorescence analysis revealed significant reduction in global αPHH3 + cells in mutant versus control embryos. Collectively, these analyses identify a dramatic impact of ectodermal loss of AP-2α and AP-2β on chromatin accessibility and gene expression of major WNT-signaling components. These changes in the ectoderm correlate with reduced proliferation of the underlying mesenchyme. In addition, these findings highlight a graded response caused by loss of three or more *Tfap2* alleles within the ectoderm with the presence of one functional allele of *Tfap2a* enabling some expression of critical regulatory genes, but loss of all four *Tfap2a/b* alleles resulting in more drastic reductions.

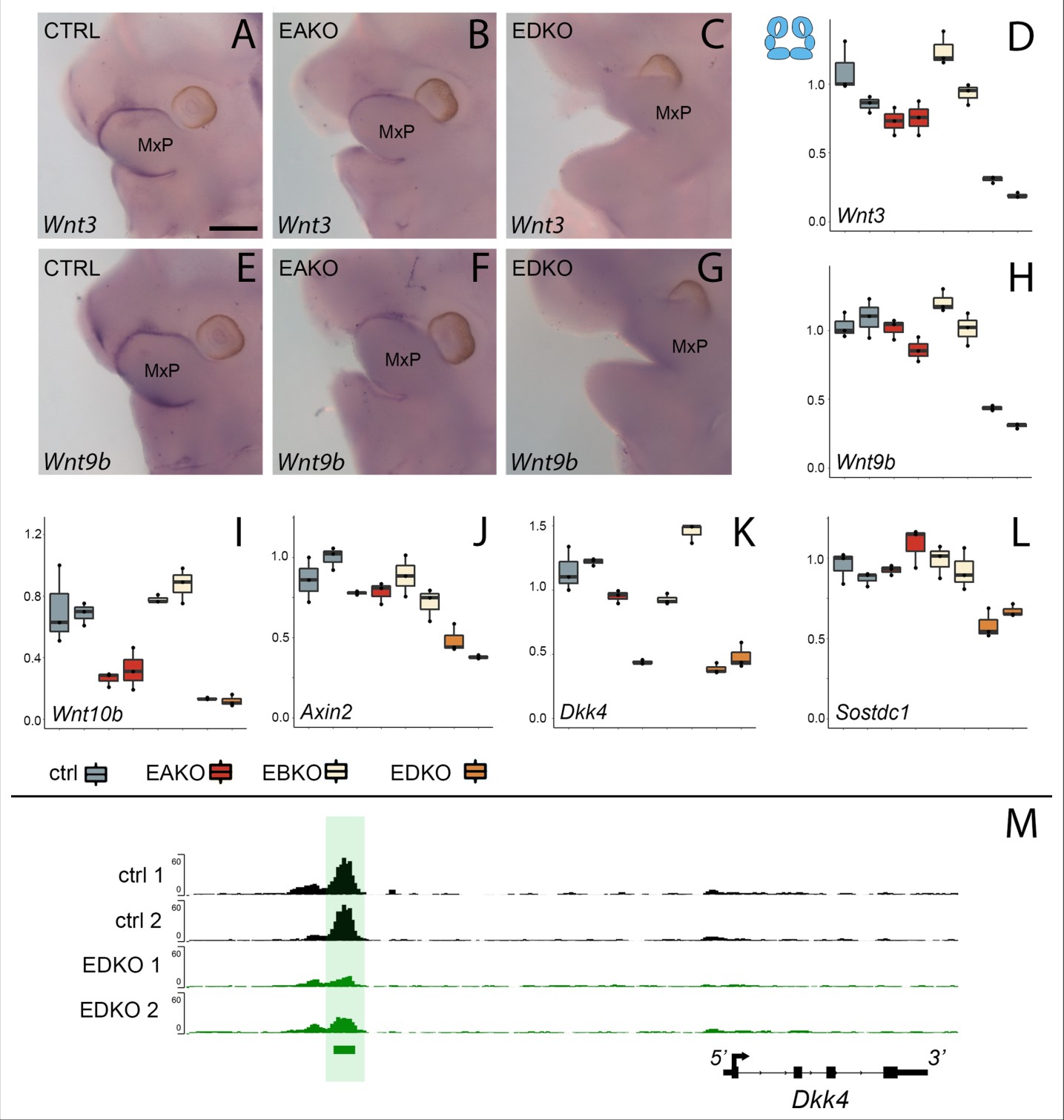

**Figure 4.** WNT-pathway related gene expression changes at E11.5 correlate with *Tfap2* gene dosage. (**A–D**). Analysis of *Wnt3* expression. (**A–C**) Lateral facial views of whole mount in situ hybridization analyses of E11.5 control (**A**), EAKO (**B**), and EDKO (**C**) embryos stained for *Wnt3*. (**D**) Quantitative RT-PCR analysis of *Wnt3* expression for biological duplicates of control (grey), EAKO (red), or EBKO (yellow) and EDKO (orange) samples. The boxplots represent technical triplicates, including upper, lower, and median values. Note, RNA was derived from whole facial prominences that is, ectoderm and mesenchyme, as shown in schematic at top left of (**D**). The Y-axis represents relative gene expression normalized to ß-actin. (**E–H**) Panels show equivalent whole mount and qRT-PCR analyses to (**A–D**) for *Wnt9b* expression. (**I–L**) Quantitative RT-PCR analysis for *Wnt10b* (**I**), *Axin2* (**J**), *Dkk4* (**K**) and *Sostdc1* (**L**) as in panel (**D**). (**M**) IGV screenshot showing tracks for ATAC-seq analysis in control (top two tracks, black, ctrl 1 and ctrl 2) or EDKO

*Figure 4 continued on next page*

*Figure 4 continued*

(bottom two tracks, green, EDKO 1 and EDKO 2), and regions of significant difference between the two genotypes (green bar). An 'AP-2-dependent' nucleosome-free region is highlighted in green ~6 kb upstream of the 4 kb mouse *Dkk4* transcription unit. MxP, maxillary prominence. Ctrl embryos are *Tfap2a^flox/+^; Tfap2b^flox/+^*, EAKO embryos are Crect; *Tfap2a^flox/flox^; Tfap2b^flox/+^*, EBKO embryos are Crect; *Tfap2a^flox/+^; Tfap2b^flox/flox^*, and EDKO embryos are Crect; *Tfap2a^flox/flox^; Tfap2b^flox/flox^*. A minimum of three embryos per genotype were used for in situ analysis, while real-time PCR was conducted with two biological replicates (each with technical triplicates). Scale bar = 500 µM.

The online version of this article includes the following figure supplement(s) for figure 4:

**Figure supplement 1.** Co-expression analysis between *Tfap2a* and multiple WNT-signaling components.

**Figure supplement 2.** IGV browser screenshot of ATAC-seq tracks at the *Sostdc1* locus.

**Figure supplement 3.** IGV browser screenshot of ATAC-seq tracks near the *Axin2* locus.

**Figure supplement 4.** Bar-charts summarizing real-time RT-PCR analysis of cDNA generated from RNA collected from E11.5 craniofacial mesenchyme (mes) or surface ectoderm (ect) of a control or EDKO (mut) sample.

**Figure supplement 5.** Quantification of cell proliferation in the facial prominences of control and EDKO embryos.

## A graded response in gross craniofacial development results from different *Tfap2a* and *Tfap2b* allelic combinations in the surface ectoderm

The graded changes in WNT pathway gene expression observed at E11.5 EBKO, EAKO, and EDKO embryos suggested that the loss of different allelic combinations of *Tfap2a* and *Tfap2b* in the facial ectoderm might also have functional consequences for facial development. After determining that certain allelic combinations did not survive postnatally, we found that at E18.5, EBKO embryos (*Figure 5C and C'*) were indistinguishable from controls (*Figure 5A and A'*), whereas EAKO (*Figure 5B and B'*) and EDKO (*Figure 5D and D'*) embryos displayed substantial defects. EAKO embryos exhibited bilateral facial clefting, a cleft palate, a cleft hypoplastic mandible, bifid tongue, hypoplastic and low-set pinna, and a partial ventral body-wall closure defect (*Figure 5B and B'*). These phenotypes were exacerbated in EDKO embryos, with most craniofacial structures severely malformed (*Figure 5D and D'*), displaying a complete failure of the facial prominences to grow towards the midline, with the maxilla and mandible growing out laterally from the oral cavity, resulting in a mandibular and palatal cleft, consistent with the morphological defects observed at earlier time points (*Figure 1*). Similarly, structures derived from the MNP and LNP failed to fuse with each other or the maxilla, instead growing dorsally, resulting in exposure of the developing nasal cavity (*Figure 5D and D'*). External pinnae were notably absent and there was also microphthalmia (*Figure 5D and D'*). Compared to the EAKO mutants, EDKO embryos also had a more severe ventral body wall closure defect, with an open thorax (*Figure 5D'*). A small percentage of EDKO mutants also had a failure of dorsal neural tube closure, resulting in exencephaly (data not shown). Finally, EDKO mutants also displayed an apparent thinning of the epidermal layer, resulting in tissue transparency, most obvious around the lateral portions of the neck (*Figure 5D*). Collectively, these findings reveal that functional redundancy exists between AP-2α and AP-2β within the ectoderm lineage—most notably in the context of facial morphogenesis. Furthermore, these results indicate that AP-2α has the most potent TF activity since mice lacking *Tfap2b*, but containing one functional copy of *Tfap2a*, can still undergo normal facial development, whereas the reverse results in orofacial clefting.

## Disruption of neural crest derived craniofacial bone and cartilage elements in EDKO mutants

To further assess the effect of loss of *Tfap2a* and *Tfap2b* within the facial ectoderm, E18.5 embryos were processed by alizarin red and alcian blue staining, revealing bone and cartilage elements, respectively (*Figure 6*). The craniofacial skeleton can be grouped into three structural units: the viscerocranium (comprising solely NCC derived facial elements); neurocranium (calvaria/skull vault); and chondrocranium—the latter two units having both a NCC and mesoderm origin reviewed in *Minoux and Rijli, 2010*. Control and EBKO embryos displayed the typical NC-derived craniofacial elements (*Figure 6A, D, G and J*, and not shown) whereas both EAKO and EDKO embryos demonstrated major disruption to several of these skeletal structures. First, in EAKO skeletons (*Figure 6B, E, H and K*), the most substantially affected structures included a shortened, cleft mandible, hypoplastic development

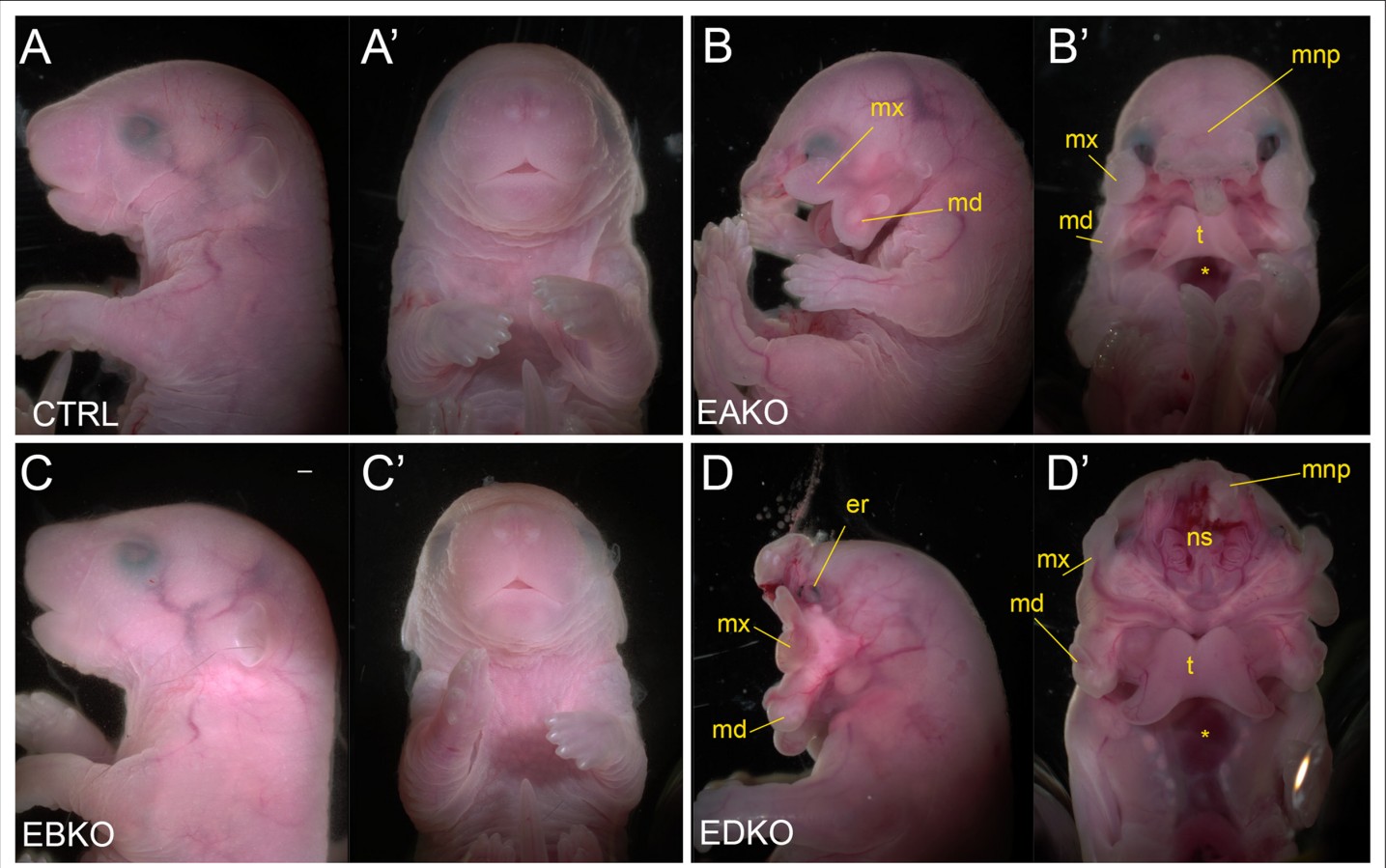

**Figure 5.** Gross morphological phenotypes of E18.5 control, EAKO, EBKO, and EDKO mutants. Lateral (**A–D**) or ventral (**A'-D'**) views of an E18.5 control (**A, A'**), EAKO (**B, B'**), EBKO (**C, C'**), or EDKO (**D, D'**) embryo. Abbreviations: md, mandible; mnp, medial nasal prominence; mx, maxillary prominence; er, eye remnant; ns, nasal septum; t, tongue. Asterisks in B' and D' indicates ventral body wall closure defect. Ctrl embryos are *Tfap2a^flox/+*; *Tfap2b^flox/+*, EAKO embryos are Crect; *Tfap2a^flox/null*; *Tfap2b^flox/+*, EBKO embryos are Crect; *Tfap2a^flox/+*; *Tfap2b^flox/null*, and EDKO embryos are Crect; *Tfap2a^flox/null*; *Tfap2b^flox/null*. A minimum of at least three embryos per genotype were examined. Mandibular clefting and failure of facial fusion was fully penetrant in EDKO embryos. Scale bar = 500 µM.

of the maxillary, nasal, lamina obturans, and palatine bones (consistent with the bilateral facial clefts and clefting of the secondary palate), a slightly hypoplastic frontal bone, and missing tympanic bones. The premaxillary bone developed anteriorly into a long bony element protruding at the front of the face, presumably due to the absence of constraints imposed by fusion to the maxilla (*Figure 6B*)—a feature commonly observed in humans with orofacial clefting (*Nyberg et al., 1993*). In addition, isolation of the mandible revealed disruption to the patterning of the proximal end, including the normally well-defined condyles seen in control embryos (*Figure 6J and K*). These skeletal defects were even more pronounced in EDKO mutants, with some additional features observed that were not seen in EAKO preparations as discussed further below. Thus, several NC derived bones that were hypoplastic in EAKO mutants were virtually absent in the EDKO mutants, including the squamosal, jugal, palatine, and lamina obturans (*Figure 6C, F, I*). Like EAKO mutants, the tympanic bones were absent, the frontal bone hypoplastic, and the premaxillary bone protruding in EDKO mutants, although this latter process grew mediodorsally reflecting the more extreme outward growth of the facial prominences in the latter genotype. Both the mandible and maxillary bones, comprising the lower and upper-jaw, respectively, were more severely impacted in EDKO mutants, including a loss of the primary and secondary jaw joints, resulting in syngnathia (*Figure 6C*). Like EAKO mutants, isolation of the mandible in EDKO mutants revealed a major loss of proximal condylar identity that was exacerbated by fusion with upper-jaw components (*Figure 6L*). Also, in contrast to EAKO embryos, the oral/aboral axis of the mandible was disrupted, resulting in a less pronounced tapering at the distal end (*Figure 6L*). To

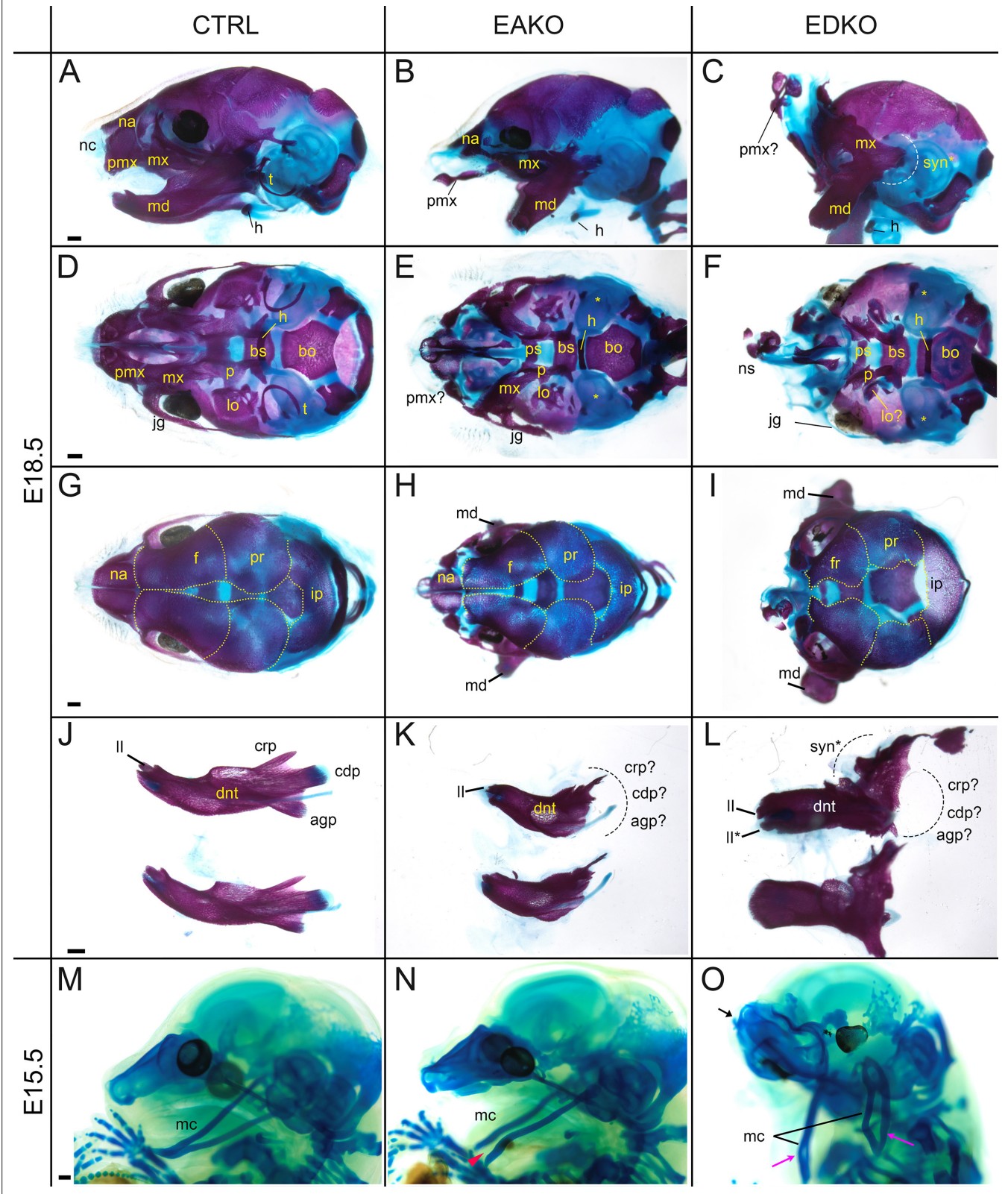

**Figure 6.** Craniofacial skeleton and chondrocranium defects vary with *Tfap2* gene dosage. (**A–L**) E18.5 alizarin red and alcian blue stained craniofacial elements. Lateral (**A–C**), ventral (**D–F**), dorsal (**G–I**) views of the craniofacial skeleton, and lateral views of the left and right hemi-mandibles in isolation (**J–L**) in control (**A, D, G, J**), EAKO (**B, E, H, K**), and EDKO (**C, F, I, L**) embryos. Note that the mandibles have been removed in (**D–F**) for clearer visualization of the cranial base, and the calvaria are outlined with yellow dashed lines in (**G–I**). The white dashed line in (**C**) highlights fusion of the upper

*Figure 6 continued on next page*

*Figure 6 continued*

and lower jaw (syngnathia), also indicated by the black dashed lines in (**L**). (**M–O**) E15.5 alcian blue stained chondrocraniums from a control (**M**), EAKO (**N**), or EDKO (**O**) embryo. A cleft Meckel's cartilage is highlighted by the red arrowhead in (**N**) or by black lines in (**O**). Note, Meckel's cartilage is also duplicated (pink arrows) along the proximodistal axis of the lower jaw in (**O**) and upturned nasal cartilages are highlighted by the black arrow. Abbreviations: agp, angular process; bs, basisphenoid; bo, basioccipital; cdp, condylar process; crp, coronoid process; dnt, dentary; f, frontal; h, hyoid; ii, inferior incisor; ii*, duplicated incisor; ip, interparietal; jg, jugal; lo, lamina obturans; mc, Meckel's cartilage; md, mandible; mx, maxillary; na, nasal; nc, nasal cartilage; ns, nasal septum; p, palatine; pmx, premaxillary; pr, parietal; ps, presphenoid; syn*, syngnathia; t, tympanic ring;? indicates possible identity of dysmorphic structure; * in (**E, F**) indicates missing tympanic ring. Ctrl embryos are *Tfap2a*$^{flox/+}$; *Tfap2b*$^{flox/+}$, EAKO embryos are Crect; *Tfap2a*$^{flox/null}$; *Tfap2b*$^{flox/+}$, and EDKO embryos are Crect; *Tfap2a*$^{flox/null}$; *Tfap2b*$^{flox/null}$. A minimum of at least three cranial skeletons per genotype were examined. Scale bar = 500 μM.

further investigate these unique features, we subsequently stained the chondrocranium of control, EAKO, and EDKO embryos at E15.5 with alcian blue (*Figure 6M–O*). Notably, this analysis revealed that EDKO mutants displayed a duplicated Meckel's cartilage along the length of the proximal-distal axis of the mandible, a feature not observed in other genotypes, and consistent with a duplication of the mandible along the oral/aboral axis (*Figure 6M–O*).

In summary, skeletal analysis indicated that the NC derived elements in the craniofacial skeleton were most exquisitely sensitive to loss of AP-2α and AP-2β from the surface ectoderm. In contrast, mesoderm derived components, such as the basioccipital of the cranial base, appeared less affected in EAKO and EDKO mutants (*Figure 6D–F*). These findings are consistent with AP-2 expression in the ectoderm affecting short range signaling to the adjacent NCC mesenchyme to control growth and morphogenesis.

## RNA-Seq analysis of E10.5 EDKO mutants reveals early disruption of WNT signaling components along with reciprocal mesenchymal perturbations

To obtain a more global assessment of the gene expression changes in the ectoderm and how they impact the underlying mesenchyme, we performed RNAseq analysis of the whole face at E10.5 for both control and EDKO mice (*Figure 7A*). This timepoint was chosen to detect primary changes in gene expression before major morphological differences were apparent in the mutants. Three biological replicates of each genotype were processed and the read data for each gene are summarized in *Supplementary file 3*. An initial assessment of the data was made by examination of a list of ~240 genes that satisfied a 1.5 fold cut-off in gene expression difference between controls and mutants, and which had consistent and measurable expression changes when viewed on the IGV browser (*Supplementary file 3*). This manually curated list revealed that multiple genes down-regulated in the mutant were associated with development and function of the ectoderm (*Table 1*). Notably, there was reduction in *Krt5*, *Krt14*, and *Krt15* expression, as well as for several genes associated with the periderm, balanced by a rise in *Krt8* and *Krt18* transcripts, indicating a delay or inhibition of normal stratification. Further, mRNAs for TFs associated with epidermal development, particularly *Trp63*, *Grhl3*, and *Foxi2*, were also reduced in the mutant (*Supplementary file 3*). Other notable changes occured in signaling molecules associated with the WNT pathway, with CXCL factors and to a lesser extent with genes involved in NOTCH, EDN, and FGF.

signaling (*Supplementary file 3*). Prominent up-regulated genes included *Lin28a* and *Cdkn1a*, which correlate with the reduced expression of genes for ectodermal differentiation and the inhibition of growth noted by more limited α-PHH3 + stained cells in the mutants (*Figure 4—figure supplement 5*).

Many of the genes we had identified had an ectodermal connection even though such genes are underrepresented in the analysis of whole prominence tissue. We therefore adopted a second approach to help distinguish the relevant tissue specific expression differences. Here, we focused on a group of 711 genes that satisfied a 1.2 fold-change and Q < 0.05 cut off between control and EDKO samples (*Figure 7B and C*, and *Supplementary file 3*). Of these, 365 were down-regulated and 346 upregulated, with no statistically significant difference between fold-change of up and down-regulated genes (*Figure 7B and D*). We next employed published gene expression levels for both the ectoderm and mesenchyme of control E10.5 wild-type embryos (*Supplementary file 3*) to distinguish the relevant tissue specific expression differences (*Hooper et al., 2020*). Of the 711 genes that were

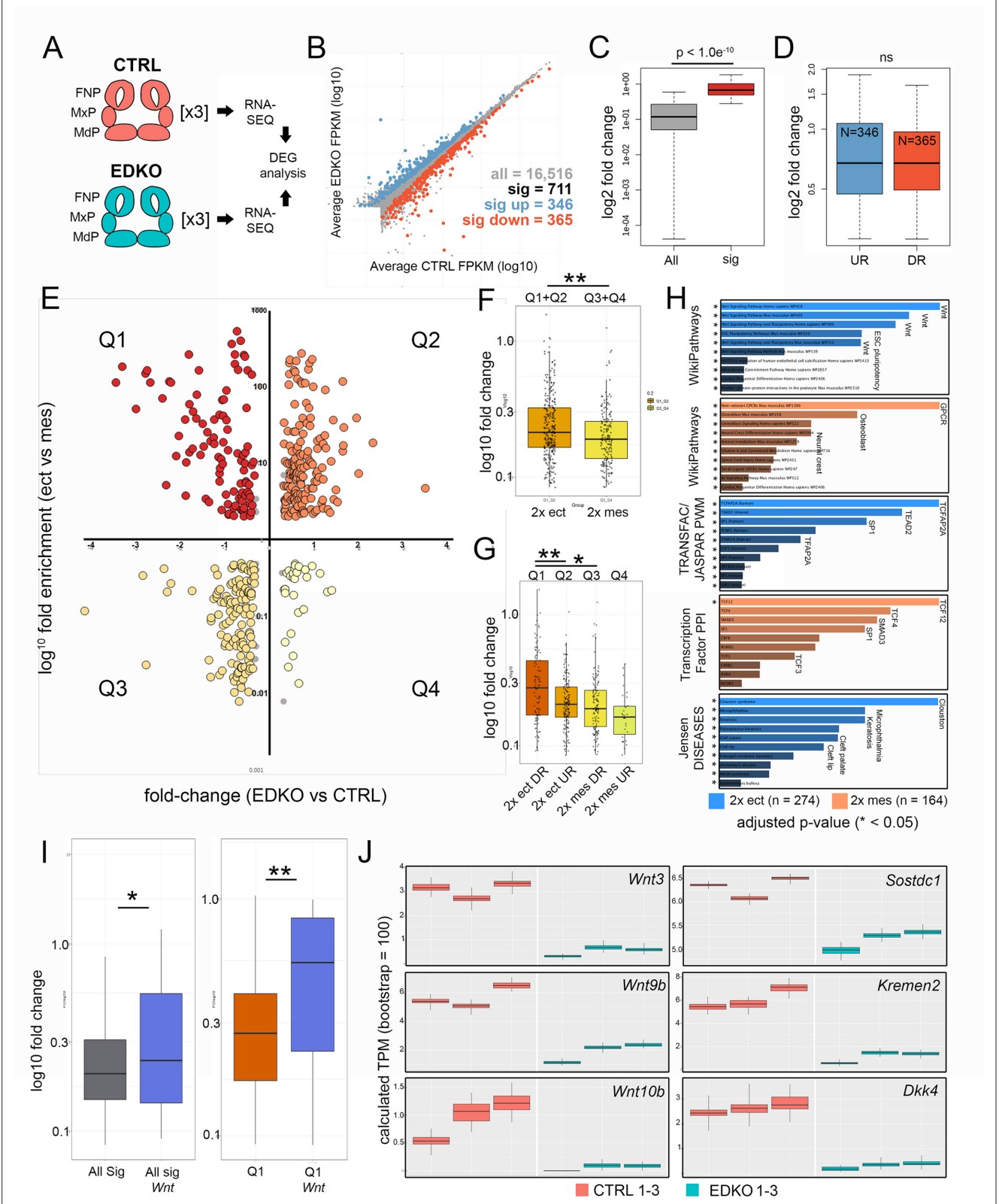

**Figure 7.** RNA-seq analysis of E10.5 control and EDKO mutant craniofacial prominences. (**A**) Schematic depicting regions isolated and general workflow for RNA-seq analysis. (**B**) Scatterplot of gene mean expression values (FPKM) for control (X-axis) and EDKO mutant (Y-axis) samples, blue or orange dots representing genes significantly upregulated or down-regulated in mutants versus controls, respectively. (**C**) Boxplot of mean fold-change values (mutant versus control) for all expressed genes (grey) or those that were significantly altered (red). (**D**) Boxplot of mean gene expression fold-change

*Figure 7 continued on next page*

*Figure 7 continued*

values (mutant versus control) for down-regulated (orange) or up-regulated (blue) genes. (**E**) Scatterplot of mean gene expression fold-change between mutant and control samples (X-axis) and mean gene expression fold-change between craniofacial ectoderm and mesenchyme (Y-axis). (**F**) Boxplot of mean gene expression fold-change values (mutant versus control) for 'ectoderm enriched' (orange) or 'mesenchyme enriched' (yellow) genes. (**G**) As in (**F**) but further subset into each quadrant. (**H**) Gene-set enrichment analysis (using ENRICHR) for 'AP-2-dependent' ectoderm (blue) or mesenchyme (orange) enriched genes. (**I**) Boxplots of mean gene expression fold-change values (mutant versus control) for all significantly altered genes (grey) versus those found specifically in the WNT-pathway (blue) or all significantly down-regulated ectoderm genes (Q1 genes, red) versus WNT-pathway associated genes down-regulated in the ectoderm (Q1 Wnt, blue). (**J**) RNA-seq based, computed gene expression values (TPM) for a subset of WNT-related genes, shown as biological triplicates in control (salmon) or EDKO mutant (teal). For all boxplots, the median is indicated by the horizontal line, 75th and 25th percentiles by the limits of the box, and the largest or smallest value within 1.5 times the interquartile range by the lines. A standard two-tailed t-test was conducted to calculate significance in C, D, F, G, and I (* = p-value < 0.05; ** = p-value < 0.005). Abbreviations: DEG, differentially expressed genes; DR, down-regulated; FNP, nasal processes; MdP, mandibular prominence; MxP, maxillary prominence; ns, not significant; TPM, transcripts per million; UR, up-regulated. Samples used for RNA-seq analysis included, 2 *Tfap2a*flox/+; *Tfap2b*flox/+ and 1 *Tfap2a*flox/+; *Tfap2b*flox/null control embryos and 3 Crect; *Tfap2a*flox/null; *Tfap2b*flox/null EDKO embryos.

The online version of this article includes the following figure supplement(s) for figure 7:

**Figure supplement 1.** Scatterplot as described for *Figure 7E*.

**Figure supplement 2.** Real-time RT-PCR (**A, D, G**) or in situ hybridization (**B, C, E, F**) of various Wnt pathway components in E10.5 control (ctrl), EAKO, or EDKO embryos.

**Figure supplement 3.** Bar-charts summarizing real-time RT-PCR analysis of cDNA generated from RNA collected from E11.5 craniofacial mesenchyme of a control or EDKO sample.

differentially expressed, 438 showed >2 fold enrichment between control tissue layers (i.e. either higher in ectoderm or higher in mesenchyme). We then used this information (*Figure 7E* Y-axis), alongside the relative change in expression between controls and mutants (*Figure 7E* X-axis), to stratify the differentially expressed genes into four major groups (Q1-4, *Supplementary file 3*). Specifically, we identified genes with preferential expression in the control ectoderm that were 'down-regulated' (*Figure 7E* and Q1, N = 103) or 'up-regulated' in mutants (*Figure 7E* and Q2, N = 171) and likewise for the mesenchyme 'down-regulated mesenchyme' (*Figure 7E* and Q3, N = 133) and 'up-regulated mesenchyme' (*Figure 7E* and Q4, N = 31). Statistical analysis of the fold-change between quadrants identified a significantly greater magnitude of fold-change in ectoderm vs. mesenchyme (*Figure 7F*) most likely due to down-regulated ectodermal genes (i.e. Q1) vs. all other quadrants (*Figure 7G*). These data suggest that, although representing a smaller fraction of the entire tissue sampled, larger changes in gene expression were within the ectoderm lineage of E10.5 mutants.

To address further how the individual genes affected in mutant vs. control embryos fit within larger biological processes and developmental systems, we utilized Enrichr (*Chen et al., 2013*; *Kuleshov et al., 2016*) along with our stratified gene lists (*Figure 7H*, *Supplementary file 3*). First, using genes differentially expressed within the ectoderm (Q1 and Q2, N = 274) we identified the most over-represented pathway was 'WNT-signaling', which occurred in four of the top five categories (*Figure 7H*)—strongly supporting our ATAC-seq and targeted gene expression analysis at E11.5. In contrast, analysis of pathways over-represented in the mesenchyme differentially expressed gene list (Q3 and Q4, N = 164), identified the top pathways to include 'GPCR', 'Osteoblast', and 'Neural crest' (*Figure 7H*). Examination of over-represented TF binding sequences within the promoters of genes mis-regulated in the ectoderm identified TFAP2A as the most significant (*Figure 7H*). Further, we assessed how the expression data correlated with the ~3.1 K AP-2-dependent promoter and enhancer peaks from the ectoderm ATAC-seq results. The Q1 genes, representing 'down-regulated ectoderm' had the greatest overlap with 56/103 (~54%) genes having AP-2 dependent peaks while in contrast, Q2 had 57/171 (33%), Q3 had 30/133 (23%), and Q4 had 10/31 (32%). The higher proportion of AP-2 dependent peaks associated with Q1 strongly suggests that AP-2 directly regulates many of these genes within the facial ectoderm, including members of the WNT pathway, IRX family, and keratins (also see *Table 1*). Conversely, genes mis-regulated in the mesenchyme were shown to be significantly enriched for TCF12/4/3-interactors based on protein-protein interaction databases (*Figure 7H*) supporting a model in which genes affected within the ectoderm are more likely direct targets of AP-2, whereas those impacted in the mesenchyme are more likely to be indirect. The ectoderm Q1/Q2 gene list also highlighted annotations for orofacial clefting (*Figure 7H*)—fitting with the clefting phenotype observed in mutant embryos. Importantly, included within this list were *TRP63* and

**Table 1.** Curated list of differentially expressed genes identified in E10.5 EDKO facial prominences vs control facial prominences, with the presence or absence of associated ATAC-seq peaks based on GREAT.

| Gene category | Gene | Average expression in control | Average expression in mutant | Fold change Mutant vs Control | AP-2 dependent ATAC-seq peak |
|---|---|---|---|---|---|
| Epithelial Development and Function | Krt5 | 3.25 | 0.52 | 0.16 | Yes |
| | Bnc1 | 2.46 | 0.50 | 0.20 | No |
| | Krt15 | 3.74 | 0.81 | 0.22 | Yes |
| | Tgm1 | 0.82 | 0.23 | 0.28 | No |
| | Hr | 0.63 | 0.27 | 0.43 | No |
| | Nectin4 | 2.73 | 1.40 | 0.51 | No |
| | Krt14 | 11.40 | 6.36 | 0.56 | Yes |
| | Perp | 16.49 | 9.72 | 0.59 | Yes |
| | Grhl3 | 4.45 | 2.61 | 0.59 | Yes |
| | Trp63 | 11.09 | 7.12 | 0.64 | Yes |
| | Krt8 | 26.20 | 39.97 | 1.53 | Yes |
| | Krt18 | 33.05 | 53.98 | 1.63 | No |
| Epithelial Junction Complexes | Gjb6 | 1.48 | 0.40 | 0.27 | Yes |
| | Gjb2 | 2.65 | 0.86 | 0.32 | Yes |
| | Gjb3 | 1.86 | 0.63 | 0.34 | No |
| | Col17a1 | 0.74 | 0.26 | 0.35 | Yes |
| | Tns4 | 1.05 | 0.48 | 0.48 | No |
| Periderm | Gabrp | 1.23 | 0.03 | 0.02 | No |
| | Zfp750 | 0.85 | 0.18 | 0.21 | Yes |
| | Rhov | 0.73 | 0.17 | 0.23 | No |
| | Krt19 | 6.93 | 12.96 | 1.87 | Yes |
| Signaling | Dkk4 | 1.96 | 0.20 | 0.10 | Yes |
| | Wnt10b | 0.84 | 0.11 | 0.13 | Yes |
| | Kremen2 | 3.98 | 0.58 | 0.15 | No |
| | Wnt3 | 2.09 | 0.31 | 0.15 | Yes |
| | Cxcl14 | 18.02 | 3.45 | 0.19 | Yes |
| | Wnt10a | 0.48 | 0.12 | 0.25 | Yes |
| | Wif1 | 1.94 | 0.49 | 0.25 | Yes |
| | Wnt9b | 3.86 | 1.12 | 0.29 | Yes |
| | Sostdc1 | 8.95 | 2.76 | 0.31 | Yes |
| | Cxcl13 | 6.55 | 2.65 | 0.40 | No |
| | Ednra | 15.93 | 9.44 | 0.59 | No |
| | Dll1 | 8.96 | 13.88 | 1.55 | Yes |
| | Fgfr3 | 5.10 | 10.10 | 1.98 | Yes |

*Table 1 continued on next page*

*Table 1 continued*

| Gene category | Gene | Average expression in control | Average expression in mutant | Fold change Mutant vs Control | AP-2 dependent ATAC-seq peak |
|---|---|---|---|---|---|
| Transcription factors | Foxi2 | 1.38 | 0.02 | 0.01 | Yes |
| | Irx4 | 1.67 | 0.15 | 0.09 | Yes |
| | Gbx2 | 4.06 | 0.60 | 0.15 | Yes |
| | Osr2 | 3.34 | 0.83 | 0.25 | No |
| | Irx2 | 4.02 | 1.17 | 0.29 | Yes |
| | Lmx1b | 2.75 | 1.05 | 0.38 | Yes |
| | Twist2 | 29.64 | 13.73 | 0.46 | No |
| | Vgll3 | 2.15 | 1.01 | 0.47 | Yes |
| | Hand1 | 11.26 | 5.36 | 0.48 | Yes |
| | Irx5 | 8.40 | 4.03 | 0.48 | Yes |
| | Twist1 | 135.74 | 79.38 | 0.58 | No |
| | Msx1 | 102.04 | 61.27 | 0.60 | Yes |
| | Sox21 | 4.56 | 7.94 | 1.74 | No |
| | Pax6 | 9.36 | 29.27 | 3.13 | Yes |
| Other | Ass1 | 2.26 | 1.30 | 0.58 | No |
| | Hapln1 | 8.59 | 5.07 | 0.59 | Yes |
| | Smoc2 | 6.72 | 2.22 | 0.33 | Yes |
| | Cdkn1a | 8.46 | 15.22 | 1.80 | No |
| | Tagln | 6.85 | 13.35 | 1.95 | No |
| | Lin28a | 9.29 | 20.20 | 2.17 | No |

*GRHL3*, which have both been associated with human clefting (*Celli et al., 1999*; *Leslie et al., 2016*; *Peyrard-Janvid et al., 2014*). Both of these genes are highly enriched in the ectoderm and down-regulated >1.5 fold within EDKO mutants, relative to controls (*Supplementary file 3*). Further, our studies show both genes possess AP-2-dependent ATAC-seq peaks, while *TRP63* is also a proposed AP-2 transcriptional target in humans (*Li et al., 2019b*).

Finally, 32 out of the total 711 differentially expressed genes were related to the WNT signaling pathway, (*Figure 7B*, *Figure 7—figure supplement 1*), and their average fold-change was significantly more than the average fold-change of the remaining 679 genes ($p < 0.05$) (*Figure 7I*). This comparison was even more significant when examining genes solely within Q1 ($p < 0.005$) (*Figure 7I*). That is, WNT-pathway genes down-regulated in the ectoderm of EDKO mutants, relative to controls, were more significantly impacted than all other genes represented in Q1. Numerous WNT components—many of which were previously identified from our ATAC-seq data—including ligands (*Wnt3*, *Wnt4*, *Wnt6*, *Wnt9b*, *Wnt10a*, *Wnt10b*), WNT inhibitors (*Dkk4*, *Kremen2*, *Sostdc1*), and a WNT receptor (*Fzd10*), were represented within this list (*Supplementary file 3*). Again, examination of available scRNAseq data (*Li et al., 2019a*) shows considerable overlap between the expression of these genes and *Tfap2a* in the surface ectoderm (*Figure 4—figure supplement 1*). Consistent with these genes being expressed in the ectoderm, their read-based calculated expression levels were often low relative to mesenchymal genes but showed striking congruence between triplicates (*Figure 7J*). We note that the reduced expression observed for several of these genes at E10.5 in the RNAseq data was also observed at E11.5 by in situ and RT-PCR analysis (*Figure 4*). Furthermore, we also validated the changes seen at E10.5 for *Wnt3*, *Wnt9b*, *Kremen2*, and *Sostdc1* using a combination of RT-PCR and in situ analysis (*Figure 7—figure supplement 2*).

Although Q1 genes, assigned as ectodermal down-regulated, had the most significant changes in expression (*Figure 7G*), several other WNT-related genes were also impacted in EDKO mutants.

Specifically, additional WNT modulators (mostly repressors), *Rspo2*, *Nkd2*, *Nkd1*, *Axin2*, *Dkk2*, and *Kremen1* were also significantly down-regulated in mutant embryos (*Supplementary file 3*). Several of these genes including *Axin2*, *Kremen1*, *Dvl2*, and *Fzd10* showed notable overlap with the *Tfap2* genes in the surface ectoderm but were also present at significant levels in the underlying mesenchyme while a further set including *Wif1*, *Dkk2*, *Rspo2*, and *Nkd1* display more prominent expression in the mesenchyme than in the ectoderm (*Figure 4—figure supplement 1*). We had previously shown that *Axin2* expression was reduced in both the ectoderm and mesenchyme (*Figure 4*), and we next extended these studies to a number of these other mesenchymally expressed WNT pathway genes. RT-PCR analysis of isolated mesenchymal RNA from control and EDKO facial prominences demonstrated that *Wif1*, *Dkk2*, *Kremen1*, and *Nkd1* were also significantly reduced in expression in the mutant tissue (*Figure 7—figure supplement 3*). We speculate their down-regulation in the mesenchyme probably results from a regulatory feedback loop caused by reduced expression of *Wnt* ligands from the ectoderm. Concurrently, several *Wnt* receptors (*Fzd5*, *Fzd8*, and *Fzd9*) and related molecules (*Sfrp1*, *Sfrp2*, and *Sfrp4*) were up-regulated (*Supplementary file 3*), potentially as a response to reduced *Wnt* ligand levels. In summary, bioinformatic and molecular analyses of control and EDKO mutants identified AP-2α and AP-2β as essential, cooperative regulators of multiple signaling pathways and processes originating from the ectoderm during craniofacial development, most notably the WNT pathway.

## WNT1 over-expression partially rescues craniofacial defects in AP-2 ectoderm mutants

*Axin2* is a direct target of WNT signaling, and the *Axin2-LacZ* allele (*Lustig et al., 2002*) was incorporated into the EAKO and EDKO mutant backgrounds as a means to determine if the loss of AP-2 alleles in the ectoderm had a direct impact on WNT pathway output. In E10.5 control embryos in which *Tfap2a/Tfap2b* had not been targeted, β-gal activity was robust within all facial prominences and the second branchial arch (*Figure 8A*). In contrast, EAKO mutants displayed a reproducible drop in β-gal staining intensity throughout these regions, with the most striking disruption around the 'hinge' (intermediate) domain of BA1 (*Figure 8B*). Finally, consistent with a more exacerbated phenotype and WNT pathway perturbation, EDKO mutants showed an even more prominent drop in β-gal staining (*Figure 8C*). Notably, β-gal activity was clearly reduced in mesenchymal populations, supporting a model in which ectodermal AP-2 influences ectodermal to mesenchymal WNT signaling.

We next assessed whether elevating WNT-signaling could mitigate the craniofacial defects observed in EAKO and EDKO embryos by incorporating an allele that expresses *Wnt1* upon Cre-mediated recombination (*Carroll et al., 2005*) into our *Tfap2* allelic series. First, though, we examined how Crect-mediated *Wnt1* overexpression in the ectoderm might impact face development to assess its suitability as a rescue model (*Figure 8—figure supplement 1*). In common with controls, E12.5 Crect *Wnt1*<sup>ox</sup> embryos had completed fusion of the face to form an intact upper lip. However, there were developmental changes in that mutant animals had a more pronounced angle between the forebrain and snout than controls and there were also defects in eye formation, consistent with activation of the WNT pathway in this process (*Smith et al., 2005*). Nevertheless, based on the overall facial phenotype, we reasoned that this approach was feasible to supplement *Wnt* ligand expression in the facial ectoderm of the EAKO and EDKO mice. In this approach, the Crect transgene both inactivates any floxed *Tfap2* alleles as well as concurrently activates *Wnt1* expression in the ectoderm (*Figure 8D*). Comparison of E13.5 EAKO to EAKO/*Wnt1*<sup>ox</sup> embryos indicated that while the former (*Figure 8E*) had bilateral cleft lip and primary palate with a protruding central premaxilla (9 of 9), most of the latter (11/13) had achieved upper facial fusion, so that there was a slight midfacial notch in place of the aberrant premaxilla as well as the formation of nares (*Figure 8F and G*). Similarly, all EDKO mice (*Figure 8H*) had facial fusion defects leading to the prominent central premaxilla (9 of 9), but in EDKO/*Wnt1*<sup>ox</sup> embryos (*Figure 8I and J*) the severity of the clefting was diminished and the central premaxilla replaced with nares (7 of 7). Note that the face was still dysmorphic in the rescued embryos, possibly reflecting insufficient WNT pathway activity, novel defects resulting from ectopic *Wnt1* expression, or additional functions regulated by AP-2 beyond the WNT pathway. Nevertheless, these data indicate that supplementing the loss of ectodermal WNT ligands in EAKO and EDKO mice can rescue major aspects of upper facial clefting fitting with our model that one of the main functions of these TFs is to regulate the WNT pathway.

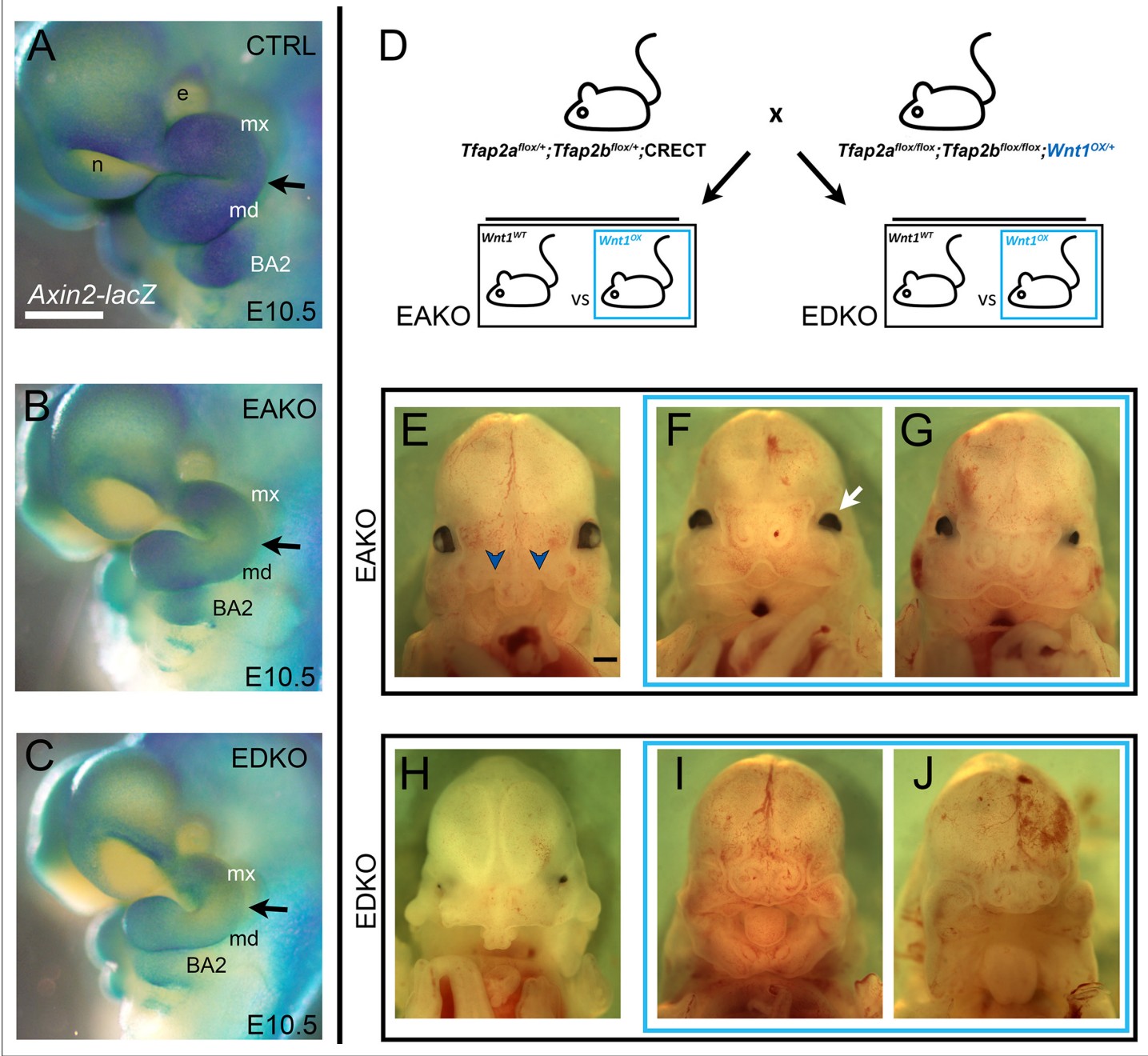

**Figure 8.** Genetic interaction between *Tfap2* and the *Wnt* pathway. (**A–C**) Lateral views of E10.5 β-galactosidase stained control (**A**), EAKO (**B**) and EDKO (**C**) embryos harboring the *Axin2-lacZ* reporter allele. The black arrow marks the position of the hinge region. (**D**) Schematic of genetic cross used to elevate Wnt1 expression levels in control, EAKO, or EDKO mutant embryos. (**E–G**) Ventral craniofacial view of E13.5 EAKO mutants that lack (**E**) or contain (**F, G**) the *Wnt1* over-expression allele. The blue chevrons indicate the bilateral cleft present in (**E**). The white arrow indicates the lack of lens development previously noted from an excess of WNT signaling (*Smith et al., 2005*). (**H–J**) Ventral craniofacial view of E13.5 EDKO mutants that lack (**H**) or contain (**I, J**) the *Wnt1* over-expression allele. Abbreviations: BA2, branchial arch 2; e, eye; md, mandibular prominence; mx, maxillary prominence; n, nasal pit. In A-C, in addition to being *Axin2-lacZ+*, Ctrl embryos are *Tfap2a^{flox/+}; Tfap2b^{flox/+}*, EAKO embryos are Crect; *Tfap2a^{flox/null}; Tfap2b^{flox/+}*, and EDKO embryos are Crect; *Tfap2a^{flox/null}; Tfap2b^{flox/null}* (n = 3/genotype).

The online version of this article includes the following figure supplement(s) for figure 8:

**Figure supplement 1.** Morphological changes associated with CRECT-mediated *Wnt1* over-expression.

## Discussion

Development of the vertebrate head requires critical regulatory interactions between various tissue layers, particularly the ectoderm and underlying neural crest derived mesenchyme. Here, we show that AP-2 transcription factors are an essential component of a mouse early embryonic ectoderm GRN directing growth and morphogenesis of the underlying facial prominence tissues. Specifically, combined conditional loss of the two most highly expressed members of the family within the ectoderm, *Tfap2a* and *Tfap2b*, results in a failure of the facial prominences to meet and fuse productively. Thus, the mandibular processes fail to meet at the midline, resulting in a wide separation between the two halves of the lower jaw and a bifid tongue. In the upper face, the maxillary, and lateral and medial nasal prominences fail to align at the lambdoid junction, resulting in an extensive bilateral cleft and significant midfacial dysmorphology. In addition to the orofacial clefting phenotypes, there was also loss of the normal hinge region between the mandible and maxilla resulting in syngnathia, and a duplication of Meckel's cartilage. Overall, the data indicate that appropriate growth, morphogenesis, and patterning of the facial prominences are all severely disrupted. The finding that AP-2α and AP-2ß work redundantly in the facial ectoderm complements studies showing that they can also work together within the cranial neural crest to control facial development (*Van Otterloo et al., 2018*). In these previous studies, the neural crest specific deletion of these factors resulted in a different type of orofacial cleft—an upper midfacial cleft—but in common with the EDKO mutants also caused syngnathia. The observations that AP-2α and AP-2ß have distinct as well as overlapping functions with both the neural crest and ectoderm for mouse facial development also inform both human facial clefting genetics and evolutionary biology. With respect to humans, *TFAP2A* mutations are associated with Branchio-Oculo-Facial Syndrome (MIM, 113620), while *TFAP2B* is mutated in Char Syndrome (MIM, 169100) (*Satoda et al., 2000*). Although both syndromes have craniofacial components to their pathology, including changes to the nasal bridge and the position of the external ears, only mutations in *TFAP2A* are associated with orofacial clefting, usually lateral. These findings support a more significant role for *TFAP2A* in influencing orofacial clefting in both mouse and human and suggest that it is the reduction of AP-2α function in the ectoderm—rather than the NCC —that is associated with this human birth defect.

In the context of chordate evolution, the prevailing hypothesis is that AP-2 has an 'ancestral' role within the non-neural ectoderm followed by an 'evolved' role within the neural crest cell lineage of vertebrates (*Meulemans and Bronner-Fraser, 2002*; *Van Otterloo et al., 2012*). The current studies further support that AP-2 proteins have critical roles within the embryonic ectoderm that have been conserved from cephalochordates and tunicates through to mammals. Alongside established roles for AP-2 transcription factors in NCCs (*Brewer et al., 2004*; *Martino et al., 2016*; *Prescott et al., 2015*; *Van Otterloo et al., 2018*), these observations raise the possibility that there may be a coordinated and complex interplay between AP-2 activity in the two tissue layers that has been conserved during evolution. The combined function of the two AP-2 factors in craniofacial development also reflects the more severe pathology resulting from the loss of more than one AP-2 gene that has been documented in other mouse developmental systems including the eye, nervous system, and skin (*Hicks et al., 2018*; *Schmidt et al., 2011*; *Wang et al., 2008*; *Zainolabidin et al., 2017*). The propensity of the AP-2 proteins to act in concert has also been observed in additional vertebrate species, particularly in the chick and zebrafish, where loss of more than one gene uncovers joint functions in neural crest, face, and melanocyte development (*Hoffman et al., 2007*; *Knight et al., 2005*; *Li and Cornell, 2007*; *Rothstein and Simoes-Costa, 2020*; *Van Otterloo et al., 2010*). Notably, in the zebrafish, previous studies have documented an interaction between AP-2α and AP-2ß during cranial NCC development (*Knight et al., 2005*). Unlike in the mouse, AP-2ß's role was confined to the surface ectoderm, based on gene expression and transplant experiments. However, more recent single-cell transcriptome profiling has identified *tfap2b* expression in the zebrafish cranial neural crest (*Mitchell et al., 2021*), suggesting features between these models (i.e. zebrafish and mouse) may be more conserved than previously thought.

The joint function of these AP-2 proteins in controlling specific aspects of gene expression presumably reflects the similar consensus sequence recognized by all family members (*Badis et al., 2009*; *Bosher et al., 1996*; *Williams and Tjian, 1991*). Thus, the absence of AP-2ß alone may not cause major developmental issues in mouse facial development due to the ability of the remaining AP-2α protein to bind and regulate shared critical genes. However, loss of both proteins would lower the

amount of functional AP-2 protein available for normal gene regulation. The hypothesis that particular levels of AP-2 are required for achieving critical thresholds of gene activity is also supported by the different phenotypes uncovered by the loss of particular *Tfap2a* and *Tfap2b* allelic combinations. Thus, while the loss of one allele of *Tfap2a*—or both alleles of *Tfap2b*—in the ectoderm is tolerated in the context of facial development, the combined loss of all three of these four alleles is not, and the phenotypes become more severe when all four alleles are defective. We note that this phenomenon was also observed when these genes were targeted in the neural crest (*Van Otterloo et al., 2018*). In both the NCCs and ectoderm the role of AP-2α seemed to be more significant than AP-2ß based on the phenotypes observed—since the presence of one functional allele of *Tfap2a* resulted in less severe pathology than the converse where only a single productive *Tfap2b* allele was expressed. The reasons behind the more prominent function of AP-2α in these systems remains unclear, especially since the two proteins are over 70% identical and bind to the same DNA recognition sequences. Nevertheless, it is possible that some alterations in the mRNA and/or amino acid sequence could differentially affect stability, localization, or interaction with cofactors. Alternatively, there might be subtle differences in the timing, distribution, or levels of functional AP-2α and AP-2ß protein in these tissues that account for the changes in susceptibility to pathology. Further studies will be needed to determine how the consequence of *Tfap2a* loss is greater than *Tfap2b*.

One notable observation, though, is that no unique and irreplaceable function exists for any AP-2α/ß heterodimers in the mouse ectoderm or neural crest. This conclusion is based on the finding that loss of *Tfap2b*—a situation that would impact both AP-2ß homodimers and AP-2α/ß heterodimers—does not impact facial development in these experiments. Finally, the sensitivity of facial development to changes in the allelic dosage of the AP-2 proteins makes this gene family a potential contributor to the evolution of facial shape. Indeed, this conjecture is supported by studies comparing genetic and morphological changes in different threespine stickleback (*Erickson et al., 2018*) and Arctic charr (*Ahi et al., 2015*) populations adapted to diverse environmental conditions, as well as by comparative studies of IPS-derived cranial neural crest cells from human and chimp, which suggest that changes in AP-2 expression and/or gene targets correlate with facial shape changes (*Prescott et al., 2015*).

The severe EDKO phenotypes also indicated that the presence or absence of these AP-2 transcription factors in the ectoderm must have a profound influence on chromatin dynamics and gene expression. To probe this in depth, ATAC-seq, H3K4me3 ChIP-seq, and RNAseq were performed on control samples derived from the embryonic mouse face, to correlate respectively chromatin accessibility, active promoter marks, and gene expression. These datasets revealed chromatin signatures that were tissue generic as well as a subset that were specific for the E11.5 facial ectoderm with the latter enriched for P53/P63/P73, AP-2, TEAD, GRHL, and PBX binding motifs. In this respect binding motifs for P53, AP-2, and TEAD family members have previously been found associated with ectodermal-specific gene regulatory pathways in embryonic skin (*Fan et al., 2018*; *Wang et al., 2006*; *Wang et al., 2008*; *Yuan et al., 2020*). Notably, our studies extend and refine the previous genome-wide analysis of embryonic skin conducted by *Fan et al., 2018* by focusing on the E11.5 facial ectoderm, enabling the detection of additional binding motifs for GRHL and PBX TF family members that are critical craniofacial patterning genes. We complemented the analysis of control samples by performing equivalent ATAC-seq and RNAseq studies on EDKO facial ectoderm or whole facial prominences, respectively. Deletion of these two transcription factors led to a significant (5%) genome-wide loss of chromatin accessibility that was centered on AP-2 consensus motifs, particularly in potential distal enhancer elements. Despite changes in the accessibility of AP-2 binding sites, motifs for P53 and TEAD family members were still highly enriched in the EDKO mutant samples. These observations provide support for previous ATAC-seq analysis of deltaNp63 mutants which hypothesized that the AP-2 and p63 programs may function independently at the protein level to regulate chromatin accessibility in embryonic ectoderm (*Fan et al., 2018*). Many of the genes linked to AP-2 binding motifs were associated with annotations aligned to skin development, such as keratins, cadherins, and gap junction proteins. However, further analysis of the gene list also revealed an evolutionary conserved group of AP-2-binding motifs connected with WNT-related genes, many associated with craniofacial development.

A strong link between AP-2 function, skin development, and WNT pathway expression was also detected in the RNAseq datasets. Changes between control and EDKO mutants in the expression of various keratin genes as well *Gjb6*, *Trp63*, *Grhl3*, and *Lin28a* suggest a failure or delay in appropriate

skin differentiation in the latter embryos. Further, loss of AP-2α/ß caused a significant reduction in expression of many *Wnt* ligands within the facial ectoderm including *Wnt3*, *Wnt6*, *Wnt9b*, and *Wnt10a*. Importantly, these four *Wnt* genes have been associated with human orofacial clefting (*Reynolds et al., 2019*), and alterations of *Wnt9b* also cause mouse CL/P (*Juriloff et al., 2005*; *Juriloff et al., 2006*). The reduced output of WNT signaling from the ectoderm was matched by a reduction of *Axin2-LacZ* reporter expression in the underlying mesenchyme, and there were also multiple changes in additional WNT components in the mesenchyme suggesting that loss of ectodermal AP-2 expression has significantly disrupted the function of this pathway throughout the developing face. Note that, although *Wnt* ligand expression is reduced in the ectoderm of EDKO mutants, it is not completely lost. Therefore, the facial pathology is not as severe as that observed with the ectodermal loss of *Wls*, a gene required for WNT ligand modification and secretion, in which the majority of the face is absent (*Goodnough et al., 2014*). The presence of teeth in the EDKO mutants (*Woodruff et al., 2021*)—although abnormal in position and number—also argues against a catastrophic loss of WNT signaling within the oral ectoderm. Further studies will be required to assess how the loss of *Tfap2a* and *Tfap2b* in the ectoderm affects other structures that require ectodermal WNT function, such as hair, whiskers, and mammary buds. With respect to facial development, additional evidence for a contribution of the WNT signaling pathway to the AP-2 mutant phenotype was obtained by overexpressing *Wnt1* in the EDKO mutant background, which resulted in a significant rescue of the facial dysmorphology and clefting. A previous study also indicated that ectopic *Wnt1* expression could rescue CL/P caused by loss of PBX expression (*Ferretti et al., 2011*), suggesting either that reduced WNT signaling is a common pathogenic mechanism for clefting or that facial growth stimulated by excess WNT signaling can mitigate the defects in juxtaposition and fusion of the facial prominences. Note that normal facial morphology was not fully recapitulated in the EDKO rescue experiments, possibly reflecting that the timing and level of *Wnt1* expression was not optimal. We also note that *Wnt1* overexpression did not affect clefting of the lower jaw in EAKO and EDKO mice, nor is the expression of genes previously linked with syngnathia and jaw-joint defects, such as *Foxc1*, *Six1*, *Bmp4*, *Fgf8*, or *Dlx* (*Jeong et al., 2008*; *Inman et al., 2013*; *He et al., 2014*; *Achilleos and Trainor, 2015*; *Liu et al., 2019*; *Abe et al., 2021*) family members significantly changed in the EDKO model. Therefore, further analysis will be required to uncover the role of ectodermal *Tfap2* expression in development of the syngnathia and other mandible defects. Indeed, our data shows that AP-2 directs additional ectodermal programs that also contribute to face formation—including IRX and IRF TF expression—as well as CXCL, EDN, FGF, and NOTCH signaling that presumably contribute to the overall pathology of the EAKO and EDKO mice. Importantly, *IRF6*—a gene involved in orofacial clefting (*Kondo et al., 2002*; *Zucchero et al., 2004*)—has previously been identified as a critical AP-2 target. Studies in human have shown that a polymorphism in an upstream enhancer element either generates or disrupts a binding site for AP-2 proteins, with the latter variant increasing the risk for orofacial clefting (*Rahimov et al., 2008*). This enhancer is conserved in the mouse, and its accessibility is altered in the EDKO mutant. Further, the expression of *Irf6* is also reduced in the EDKO mutants (*Supplementary file 3*), correlating with the loss of AP-2 binding, and providing a further pathway that might contribute to the overall phenotype. In summary, the combination of ATAC-seq, ChIP-Seq, and expression analyses highlight critical genes that are impacted by loss of AP-2 transcription factors. These data greatly expand our understanding of the gene regulatory circuits occurring in the ectoderm that regulate facial development and underscore a critical role for AP-2α and AP-2ß in controlling appropriate genome access as well as gene expression.

## Materials and methods

**Key resources table**

| Reagent type (species) or resource | Designation | Source or reference | Identifiers | Additional information |
|---|---|---|---|---|
| Genetic reagent (*Mus musculus*) | *Tfap2a*^tm1Will^ | *Zhang et al., 1996* | *Tfap2a* null allele | In-house |
| Genetic reagent (*Mus musculus*) | *Tfap2a*^tm2Will/J^ | *Brewer et al., 2004* | *Tfap2a* conditional allele | In-house |

*Continued on next page*

*Continued*

| Reagent type (species) or resource | Designation | Source or reference | Identifiers | Additional information |
|---|---|---|---|---|
| Genetic reagent (*Mus musculus*) | *Tfap2b*^tm1Will | **Martino et al., 2016**; **Van Otterloo et al., 2018** | *Tfap2b* null allele | In-house |
| Genetic reagent (*Mus musculus*) | *Tfap2b*^tm2Will | **Martino et al., 2016**; **Van Otterloo et al., 2018** | *Tfap2b* conditional allele | In-house |
| Genetic reagent (*Mus musculus*) | Crect | **Schock et al., 2017** | Crect transgene allele | In-house; Cre-driver line with Cre driven by a *Tfap2a* intronic enhancer |
| Genetic reagent (*Mus musculus*) | B6.129P2-*Axin2*^tm1Wbm/J | **Lustig et al., 2002** | Axin2lacZ | Obtained from Jackson Laboratory |
| Genetic reagent (*Mus musculus*) | Gt(ROSA)26Sor^tm2(Wnt1/Gfp)Amc/J | **Carroll et al., 2005** | Wnt1^Ox | Obtained from Jackson Laboratory |
| Genetic reagent (*Mus musculus*) | Gt(ROSA)26Sor^tm1Sor | **Soriano, 1999** | r26r | Obtained from Jackson Laboratory |
| Genetic reagent (*Mus musculus*) | Gt(ROSA)26Sor^tm4(ACTB-tdTomato,-EGFP)Luo/J | **Muzumdar et al., 2007** | mT/mG | Obtained from Jackson Laboratory |
| Antibody | anti-H3K4Me3 (Rabbit, monoclonal) | Millipore, cat. #04–745 | | 2.5 µL/ChIP |
| Antibody | anti-p-Histone H3 (Rabbit, polyclonal) | sc-8656-R, Santa Cruz Biotechnology | | 1:250 dilution |
| Software, algorithm | NGmerge | **Gaspar, 2018** | | Read trimming |
| Software, algorithm | Bowtie2 | **Langmead et al., 2009** | | Mapping |
| Software, algorithm | Samtools | **Li et al., 2009** | | Format conversion |
| Software, algorithm | Genrich (v0.5) | https://github.com/jsh58/Genrich, **Gaspar, 2022** | RRID:SCR_002630 | ATAC-seq peak calling |
| Software, algorithm | Picard (v2.19) | http://broadinstitute.github.io/picard | RRID:SCR_006525 | Duplicate removal |
| Software, algorithm | deepTools | **Ramírez et al., 2016** | | Read normalization/visualization |
| Software, algorithm | GREAT algorithm (v4) | **McLean et al., 2010** | | Pathway enrichment |
| Software, algorithm | HOMER | **Heinz et al., 2010** | | Motif enrichment |
| Software, algorithm | Trim Galore! | Babraham Bioinformatics, Babraham Institute, Cambridge, UK | | Read trimming |
| Software, algorithm | HISAT2 | **Pertea et al., 2016** | | Read mapping |
| Software, algorithm | StringTie | **Pertea et al., 2016** | | RNA expression quantification |
| Software, algorithm | CuffDiff2 | **Trapnell et al., 2012** | | Differential gene expression |
| Software, algorithm | kallisto | **Bray et al., 2016** | | RNA expression quantification |
| Software, algorithm | sleuth | **Pimentel et al., 2017** | | Differential gene expression and visualization |

## Animal procedures

All experiments were conducted in accordance with all applicable guidelines and regulations, following the 'Guide for the Care and Use of Laboratory Animals of the National Institutes of Health'. The animal protocol utilized was approved by the Institutional Animal Care and Use Committee of the University of Colorado – Anschutz Medical Campus (animal protocol #14) and the University of Iowa (animal protocol #9012197). Noon on the day a copulatory plug was present was denoted as embryonic day 0.5 (E0.5). For the majority of experiments, littermate embryos were used when comparing between genotypes. Yolk sacs or tail clips were used for genotyping. DNA for PCR was extracted

using DirectPCR Lysis Reagent (Viagen Biotech) plus 10 µg/ml proteinase K (Roche), incubated overnight at 65 °C, followed by heat inactivation at 85 °C for 45 min. Samples were then used directly for PCR-based genotyping with primers (*Supplementary file 4*) at a final concentration of 200 nM using the Qiagen DNA polymerase kit, including the optional Q Buffer solution (Qiagen).

## Mouse alleles and breeding schemes

The *Tfap2a* null (*Tfap2a^tm1Will^ Zhang et al., 1996*) and conditional alleles (*Tfap2a^tm2Will/J^* [*Brewer et al., 2004*]), the *Tfap2b* null (*Tfap2b^tm1Will^*) and conditional alleles (*Tfap2b^tm2Will^* [*Martino et al., 2016*; *Van Otterloo et al., 2018*]), as well as Crect transgenic mice (*Schock et al., 2017*), have been described previously. Crect is a Cre-driver line with Cre driven by a *Tfap2a* intronic enhancer. *Axin2lacZ* (B6.129P2-*Axin2^tm1Wbm/J^*) and *Wnt1^Ox^* (*Gt(ROSA)26Sor^tm2(Wnt1/Gfp)Amc/J^*) mice (*Carroll et al., 2005*; *Lustig et al., 2002*) were obtained from Jackson Laboratory (Bar Harbor, ME). Note that the *Wnt1^ox^* allele was always introduced into the experimental embryos via the dam, to avoid premature activation of this allele in the sire as this genetic interaction was lethal. EDKO experiments were performed using mice that were either Crect; *Tfap2a^flox/flox^*; *Tfap2b^flox/flox^* or Crect; *Tfap2a^null/flox^*; *Tfap2b^null/flox^* as indicated in the text. Similarly, EBKO mice were either Crect; *Tfap2a^flox/+^*; *Tfap2b^flox/flox^* or Crect; *Tfap2a^flox/+^*; *Tfap2b^null/flox^* and EAKO mice either Crect; *Tfap2a^flox/flox^*; *Tfap2b^flox/+^* or Crect; *Tfap2a^null/flox^*; *Tfap2b^flox/+^*. We did not detect any gross morphological differences between the two types of EDKO, EAKO, or EBKO mice which differ in respect to the number of functional *Tfap2a* or *Tfap2b* alleles in tissues that do not express Crect. Although the Crect transgene is used here to target the early embryonic ectoderm, a previous report has indicated that it frequently produces a broader pattern of recombination (*Schock et al., 2017*). We used several approaches to avoid this broader expression pattern. First, the Crect transgene was always introduced into the experimental embryos via the sire to reduce global recombination sometimes seen with transmission from the female. Second, all sires were tested using reporter lines such as (Gt(ROSA)26Sor^tm1Sor^) (*Soriano, 1999*) or mT/mG, Gt(ROSA)26Sor^tm4(ACTB-tdTomato,-EGFP)Luo/J^ (*Muzumdar et al., 2007*) to ensure that they consistently produced the desired pattern of recombination before they were used to generate EAKO, EBKO, or EDKO animals (*Figure 1—figure supplement 1*). We also confirmed the overlap between the expression of *Cre*, *Tfap2a*, and *Tfap2b* in the ectoderm by mining a previously published single cell RNAseq dataset (*Li et al., 2019a*; *Figure 1—figure supplement 1*). Lastly, we determined that the Crect transgene was highly efficient at targeting the *Tfap2a* and *Tfap2b* loci based upon RNA expression and genomic recombination associated with these genes in *Cre* expressing cells (*Figure 1—figure supplement 2*).

## Tissue preparation for ATAC-Seq

For ATAC-seq analysis, E11.5 embryos were dissected into ice-cold PBS and associated yolk sacs used for rapid genotyping using the Extract-N-Amp Tissue PCR kit as recommended by the manufacturer (Sigma). During genotyping, the facial prominences were carefully removed from individual embryos using a pair of insulin syringes and placed in a 24-well plate with 1 mL of 1 mg/ml Dispase II (in PBS). The samples were incubated with rocking at 37 °C for 30–40 min and then the facial ectoderm carefully dissected away from the mesenchyme into ice-cold PBS, as described (*Li and Williams, 2013*). Facial ectoderm was then centrifuged at 4 °C, 500 g, for 3 min in a 1.5 mL Eppendorf tube, washed 1 x with ice-cold PBS, and then centrifuged again. Following resuspension in 750 µL of 0.25% trypsin-EDTA, samples were incubated at 37 °C for 15 min with gentle agitation. Following addition of 750 µL of DMEM with 10% FBS to inhibit further digestion, cells were dissociated by pipetting up and down multiple times with wide orifice pipette tips. Cells were subsequently spun at 300 g for 5 min and washed with PBS containing 0.4% BSA, and this step was repeated twice. Finally, the cell pellet was resuspended with 50 µL of PBS and the density of the single-cell suspension quantified on a hematocytometer.

## ATAC-Seq transposition, library preparation, and sequencing

Following genotype analysis of embryos used for facial ectoderm isolation, EDKO (Crect; *Tfap2a^flox/flox^*; *Tfap2b^flox/flox^*) and control littermate samples lacking Crect (*Tfap2a^flox/+^*; *Tfap2b^flox/+^*) were used for the ATAC-seq protocol, largely following procedures previously described (*Buenrostro et al., 2013*; *Buenrostro et al., 2015*; *Corces et al., 2017*). Briefly, 50,000 cells from each sample were pelleted at 500 g for 5 minutes at 4 °C. The pellet was then resuspended in 50 µL of cold lysis buffer (10 mM

Tris-HCl, pH 7.5; 10 mM NaCl; 3 mM MgCl$_2$; 0.1% NP-40, 0.1% Tween-20; 0.01% Digitonin) by gently pipetting ~4 times to release the nuclei which were then incubated on ice for 3 min. The sample was next spun at 500 g for 20 min at 4 °C and the pelleted nuclei resuspended in Tagmentation mix (e.g. 25 µL 2 x Nextera TD Buffer, 2.5 µL Nextera TD Enzyme, 0.1% Tween-20, 0.01% Digitonin, up to 50 µL with nuclease-free water) and placed at 37 °C for 30 min in a thermocycler. Following transposition, samples were purified using the QIAGEN minElute PCR Purification Kit (Qiagen) and eluted with 11 µL of supplied Elution Buffer. Transposed DNA was next indexed with a unique barcoded sequence and amplified prior to sequencing. Briefly, 10 µL of transposed DNA was mixed with the Nextera Ad1 PCR primer as well as a unique Nextera PCR primer (e.g. Ad2.x) and NEBNext HighFidelity 2 x PCR Master Mix. Samples were then amplified using the following cycling parameters: [72 °C, 5 min], [98 °C, 30 sec], [98 °C, 10 sec; 63 °C, 30 sec; 72 °C, 1 min (repeat 10–12 cycles)]. Following cycle 5, an aliquot of sample was removed for Sybr-green based quantification to determine the number of remaining cycles required to reach adequate amounts for sequencing without introducing over-amplification artifacts due to library saturation. Following indexing and amplification, samples were purified using two rounds of AmpureXP bead-based size selection. Library purity, integrity, and size were then confirmed using High Sensitivity D1000 ScreenTape and subsequently sequenced using the Illumina NovaSEQ6000 platform and 150 bp paired-end reads to a depth of ~75 × 10$^6$ reads per sample, carried out by the University of Colorado, Anschutz Medical Campus, Genomics and Microarray Core.

## H3K4me3 histone ChIP

For H3K4me3 based histone ChIP-seq analysis, craniofacial ectoderm was first isolated from E10.5 and E11.5 wild-type mouse embryos, as previously described (*Li and Williams, 2013*). Once isolated and pooled, tissue/chromatin was crosslinked with 1% formaldehyde at RT for 10 min. Following cross-linking, reactions were quenched using 0.125 M glycine, followed by multiple PBS washes. Samples were subsequently frozen in liquid nitrogen and stored at –80 °C. Once ~5 mg of tissue was collected per stage (e.g. E10.5, N = ~ 50 embryos; or E11.5, N = ~ 15 embryos), samples from multiple dissections, but similar stages, were pooled and combined with 300 µl of 'ChIP Nuclei Lysis buffer' (50 mM Tris-HCl, pH 8.0, 10 mM EDTA, 1% SDS), with 1 mM PMSF and 1 X proteinase inhibitor cocktail (PICT, 100 X from Thermo Scientific, Prod # 1862209). Pooled tissue was resuspended completely and subsequently incubated at RT for 10 mins. Following incubation, chromatin was fragmented using a Bioruptor (Diagenode, Cat. No. UCD-200) with the following settings: High energy, 30 s on, 30 s off, with sonication for 45 min. Following shearing, chromatin was assessed as ~100–500 bp in size. Next, a small portion of fragmented chromatin was saved as input, while the rest was diluted 1 in 5 in RIPA buffer (150 mM NaCl, 1% NP-40, 0.5% deoxycholate, 0.1% SDS, 50 mM Tris pH 8.0, 5 mM EDTA, plus PMSF and PICT) followed by the addition of 20 µl protein A/G agarose beads (Pierce, Thermo Scientific, Prod # 20423) prewashed with RIPA buffer to eliminate non-specific binding. The pre-cleaned chromatin was then incubated with 2.5 µL of monoclonal H3K4Me3 primary antibody (Millipore, cat. #04–745), while rotating at 4 °C, overnight. The following day, 20 µl protein A/G beads pre-saturated with 5 mg/ml BSA in PBS (Sigma, A-3311) were washed in RIPA buffer and subsequently added to the chromatin/antibody mix at 4 °C, rotating, for 2 hr. Samples were then washed twice in RIPA, four times in Szak Wash (100 mM Tris HCl pH 8.5, 500 mM LiCl, 1% NP-40, 1% deoxycholate), twice more in RIPA followed by two TE washes (10 mM Tris HCl pH 8.0, 1 mM EDTA pH 8.0). Finally, the bead slurry was resuspended in 100 µl TE and the remaining bound chromatin was eluted off the beads using 200 µl 1.5 X 'Elution Buffer' (70 mM Tris HCl pH 8.0, 1 mM EDTA, 1.5% SDS) at 65 °C for 5 min. Once eluted, crosslinks were reversed by incubating ChIP'd samples and input samples at 65 °C overnight in 200 mM NaCl. Samples were then subjected to 20 µg of Proteinase K digestion at 45 °C for 1 hr and DNA subsequently extracted using a standard Phenol:Chloroform, EtOH-precipitation based approach. Purified, pelleted, DNA was then resuspended in 20 µl water.

## H3K4me3 histone ChIP-Seq library preparation and sequencing

Once purified fragments were obtained and quality and size confirmed, libraries were constructed using the Nugen ChIP Seq Library Construction Kit. Library purity, integrity, and size were then confirmed using High Sensitivity D1000 ScreenTape and subsequently sequenced using an Illumina MiSEQ platform and 50 bp single-end reads to a depth of ~25–30 × 10$^6$ reads per experimental

sample and ~10 × 10⁶ reads for input, carried out by the University of Colorado, Anschutz Medical Campus, Genomics and Microarray Core.

## Bioinformatic processing of ATAC-Seq and histone ChIP-Seq data

### ATAC-seq trimming, mapping, peak calling:

Following sequencing and demultiplexing, paired-end reads from each sample were first trimmed using NGmerge (with the adapter-removal flag specified) (*Gaspar, 2018*). Following trimming, samples were individually mapped to the Mm10 genome using Bowtie2 (*Langmead et al., 2009*) with the following settings (--very-sensitive -k 10) and converted to bam format and sorted using Samtools (*Li et al., 2009*). To find sites of 'enrichment' (i.e. peak calling) we used Genrich (https://github.com/jsh58/Genrich, *Gaspar, 2022*) with the following flags set (-j, -y, -r, -e chrM). First to identify control peaks, we used the two control replicate ATAC-seq alignment files—produced from Bowtie2/Samtools—as 'experimental input', with the above Genrich settings (in this approach, 'background' is based on the size of the analyzed genome, i.e., Mm10, minus mitochondrial DNA). We did a similar analysis using the two mutant replicate alignment files as 'experimental input' (rather than control)—identifying significantly enriched regions in the mutant dataset. Additionally, to compare the two datasets directly, we supplied the two control alignment files as 'experimental' while simultaneously supplying the two mutant alignment files as 'background', thus, identifying regions that were significantly enriched in controls relative to mutants. These analyses resulted in genomic coordinates of 'peaks' for each of the supplied datasets.

### H3K4me3 histone ChIP-seq trimming, mapping, and overlapping

Following sequencing, samples were demultiplexed and mapped to the Mm10 genome build using NovoAlign (Novocraft). Mapped reads were then processed for duplicate removal using the Picard suite of tools (http://broadinstitute.github.io/picard). The resulting deduplicated mapped reads were subsequently indexed using Samtools (*Li et al., 2009*) and the resulting indexed Bam files were normalized using the *bamCoverage* function in deepTools (*Ramírez et al., 2016*). The resulting normalized bigWig files were then used with the control ATAC-seq bed file (genomic coordinates of peaks), along with the *computeMatrix* function in deepTools, to generate a matrix file. This matrix was then visualized using the *plotHeatmap* function in deepTools with a K-means cluster setting of 2, identifying ATAC-seq coordinates that had high or little to no H3K4me3 enrichment.

### Multi-organ ATAC-seq dataset overlapping

First, publicly available ATAC-seq datasets were downloaded from the ENCODE consortium in bigWig file format (E11.5 heart: ENCSR820ACB; E11.5 liver: ENCSR785NEL; E11.5 hindbrain: ENCSR012YAB; E11.5 midbrain: ENCSR382RUC; E11.5 forebrain: ENCSR273UFV; E11.5 neural tube: ENCSR282YTE; E15.5 kidney: ENCSR023QZX; E15.5 intestine: ENCSR983JWA). A matrix file was then generated using all bigWig files along with the genomic coordinates obtained from the H3K4me3 clustering above (specifically the coordinates from the H3K4me3 negative cluster) using the *computeMatrix* function in deepTools. Once generated, the matrix file was then visualized using the *plotHeatmap* function, with a K-means cluster setting of 3, in deepTools.

### GREAT analysis:

To determine and plot the general distribution of sub-clusters and their genomic coordinates relative to transcriptional start site of genes, the GREAT algorithm (v4) (*McLean et al., 2010*) was used with default settings. GREAT was also utilized for identifying enriched biological pathways and gene sets within discrete sub-clusters, with the 'Association rule settings' limited to 100 kb distal in the 'Basal plus extension' setting.

### Motif enrichment analysis

For motif enrichment analysis, genomic coordinates were supplied in BED file format to the HOMER software package (*Heinz et al., 2010*), using the "findMotifsGenome.pl" program and default settings.

## Association of gene expression and ATAC-seq peaks

First, gene expression for the craniofacial ectoderm and mesenchyme, at E11.5, was calculated using our publicly available datasets profiling the facial ectoderm and mesenchyme from E10.5 through E12.5 (*Hooper et al., 2020*) (available through the Facebase Consortium website, https://www.face-base.org/, under the accession number FB00000867). Expression values for all 3 craniofacial prominences (e.g. mandibular, maxillary, frontonasal) were averaged independently for the ectoderm and mesenchyme, establishing an 'expression value' for each tissue compartment of the entire face at E11.5. Next, an 'ectoderm enrichment' value was calculated for each gene by taking the quotient of the ectoderm value divided by the mesenchyme value. Concurrently, ATAC-seq peaks from various sub-clusters were associated with a corresponding gene(s) using the GREAT algorithm and these associations were downloaded using the '*Gene - > genomic regions association table*' function in GREAT. A 'peak-associated profile' was then ascribed for each gene (i.e. the type and number of sub-cluster peaks associated with each gene), allowing the binning of genes based on this profile. Bins of genes, and their associated 'ectoderm enrichment' value were then plotted in R using the empirical cumulative distribution function (stat_ecdf) in ggplot2 and significance calculated using a Kolmogorov-Smirnov test (ks.test).

## Conservation analysis

To determine the level of conversation for AP-2-dependent genomic elements (*Figure 3E*) the phastCons60way (scores for multiple alignments of 59 vertebrate genomes to the mouse genome) dataset was downloaded from the University of California, Santa Cruz (UCSC) genome browser in bigWig format (http://hgdownload.cse.ucsc.edu/goldenpath/mm10/phastCons60way/). A matrix file was then generated using the bigWig file along with the 'AP-2 dependent' genomic coordinates using the *computeMatrix* function in deepTools. Once generated, the matrix file was then visualized using the *plotHeatmap* function, with a K-means cluster setting of 2, in deepTools.

## RNA-sequencing

For RNA-sequencing E10.5 facial prominences encompassing ectoderm and mesenchyme of the mandibular, maxillary, and nasal prominences were micro-dissected in ice cold PBS using insulin syringes and stored in RNA-later at –20 °C. Once sufficient EDKO (Crect; *Tfap2a$^{flox/null}$*; *Tfap2b$^{flox/null}$*) and control littermate samples lacking Crect (e.g. *Tfap2a$^{flox/+}$*; *Tfap2b$^{flox/+}$*) were identified for three biological replicates of each, tissue was removed from RNA-later and RNA harvested as previously described using the microRNA Purification Kit (Norgen Biotek) and following manufacturer's protocol (*Van Otterloo et al., 2018*). Following elution, mRNA was further purified using the Qiagen RNAeasy Kit according to the manufacturer's protocol. The quality of extracted mRNA was assessed using DNA Analysis ScreenTape (Agilent Technologies) prior to library production. Following validation of extracted mRNA, cDNA libraries were generated using the Illumina TruSeq Stranded mRNA Sample Prep Kit. All libraries passed quality control guidelines and were then sequenced using the Illumina HiSeq2500 platform and single-end reads (1 × 150) to a depth of ~15–25 × 10$^6$ reads per sample. To identify differentially expressed genes between control and mutant groups, we next utilized a standard bioinformatic pipeline for read filtering, mapping, gene expression quantification, and differential expression between groups (see below). Library construction and sequencing was carried out by the University of Colorado, Anschutz Medical Campus, Genomics and Microarray Core.

## Bioinformatic processing of RNA-Seq data

Raw sequencing reads were demultiplexed and fastq files subsequently processed, as previously described (*Van Otterloo et al., 2018*). Briefly, reads were trimmed using the Java software package Trim Galore! (Babraham Bioinformatics, Babraham Institute, Cambridge, UK) and subsequently mapped to the Mm10 genome using the HISAT2 software package (*Pertea et al., 2016*) (both with default settings). Following mapping, RNA expression levels were generated using StringTie (*Pertea et al., 2016*) and differential expression computed between genotypes using CuffDiff2 (*Trapnell et al., 2012*), with a significance cut-off value of Q < 0.05 (FDR-corrected p-value). As a secondary approach, particularly for plotting differential gene expression differences for specific transcripts (e.g. *Figure 7J*), quantification of transcript abundance was calculated using kallisto (*Bray et al., 2016*) and then compared and visualized using sleuth (*Pimentel et al., 2017*).

## Skeletal staining

Concurrent staining of bone and cartilage in E18.5 embryos occurred as previously described (*Van Otterloo et al., 2016*). Briefly, following euthanasia and removal of skin and viscera, embryos were first dehydrated in 95% EtOH and then for ~2 days in 100% Acetone. Embryos were then incubated in a mixture of alcian blue, alizarin red, acetic acid (5%) and 70% EtOH, at 37 °C, for ~2–3 days. Samples were then placed in 2% KOH (~1–2 days) and then 1% KOH (~1–2 days) to allow for clearing of remaining soft tissue. Final skeletal preparations were stored at 4 °C in 20% glycerol. Staining of only cartilage in E15.5 embryos occurred as previously reported (*Van Otterloo et al., 2016*). Briefly, following fixation in Bouin's at 4 °C overnight, embryos were washed with repeated changes of 70% EtOH and 0.1% NH$_4$OH until all traces of Bouin's coloration was removed. Tissue was permeabilized by two 1 hr washes in 5% acetic acid, followed by overnight incubation in a solution of methylene blue (0.05%) and acetic acid (5%). Next, embryos were washed twice with 5% acetic acid (~1 hr each wash) and then twice with 100% MeOH (~1 hr each wash). Finally, embryos were cleared with a solution consisting of one-part benzyl alcohol and two parts benzyl benzoate (BABB). A minimum of three embryos were analyzed per genotype.

## In situ hybridization

Embryos were fixed overnight in 4% PFA at 4 °C and then dehydrated through a graded series of MeOH:PBST washes and stored in 100% MeOH at –20 °C. Prior to hybridization they were rehydrated from MeOH into PBST as previously described (*Simmons et al., 2014*; *Van Otterloo et al., 2016*). Note, for some experiments, embryonic heads were bisected in a mid-sagittal plane, with either half being used with a unique anti-sense RNA probe. At a minimum, each in situ probe examined was run on three control and three EDKO mutant embryos. Antisense RNA probes were generated using a unique fragment that was cloned into a TOPO vector (Life Technologies, Grand Island, NY), using cDNA synthesized from mouse embryonic mRNA as a template. cDNA was generated using the SuperScript III First-Strand Synthesis System (Life Technologies, Grand Island, NY), as per manufacturer's instructions. The *Wnt3* probe is equivalent to nucleotides 674–1727 of NM_009521.2; *Wnt9b* to nucleotides 1158–2,195 of NM_011719; *Kremen2* probe is equivalent to nucleotides 206–832 of NM_028416. Sequence verified plasmids were linearized and antisense probes synthesized using an appropriate DNA-dependent RNA polymerase (T7/T3/SP6) and DIG RNA labeling mix (Roche, Basel, Switzerland).

## Cell proliferation analysis

To analyze cell proliferation in sectioned mouse embryos, E11.5 embryos were harvested and fixed overnight in 4% PFA at 4 °C. The following day, embryos were moved through a series of PBS and sucrose washes, followed by a mixture of sucrose and OCT. Embryos were then transferred to a plastic mold containing 100% OCT. After orientating the tissue samples in the plastic molds, the OCT 'block' was frozen on dry ice and stored at –80 °C. OCT blocks, containing control and mutant embryos, were then sectioned at 12 µM on a cryostat. Sectioned materials were stored at –80 °C. For immuno-labeling, slides which contained the frontonasal, maxillary, or mandibular prominence were brought to room temperature, washed 4 × 15 min in PBST, blocked for 1 hr in PBST plus 3% milk. Sections were then incubated overnight in primary antibody (anti-p-Histone H3, sc-8656-R, Santa Cruz Biotechnology, rabbit polyclonal) diluted 1:250 in PBST at 4 °C in a humidified chamber. Following primary antibody incubation, samples were washed twice for 10 min in PBST at room temperature, followed by a 30 min wash in PBST/3% milk. Samples were then incubated for 1 hr with a secondary antibody (goat anti-rabbit IgG, Alexa Flour 488 conjugate, ThermoFisher Scientific/Invitrogen, R37116) and DRAQ5 (Abcam, ab108410) nuclear stain, diluted 1:250 and 1:5000, respectively, in PBST. Processed samples were imaged on a Leica TCS SP5 II confocal microscope and individual images taken for visualization. After acquiring an image of each prominence, the area of interest was outlined in Image-J and immuno-positive cells within that area were counted by an independent observer—who was blinded to the sample genotype—using the *threshold* and *particle counter* function. The number of positive cells/area of the 'area of interest' (e.g. the prominence) was then calculated for sections originating from three control and three EDKO embryos. An unpaired student T-test was used to assess statistical significance between groups.

## ß-Galactosidase staining

Whole-mount ß-galactosidase staining was conducted as previously described (*Seberg et al., 2017*). Briefly, embryos were fixed for ~30 min to 1 hr at RT in PBS containing 0.25% glutaraldehyde, washed 3 × 30 min in a 'lacZ rinse buffer' followed by enzymatic detection using a chromogenic substrate (1 mg/ml X-gal) diluted in a 'lacZ staining solution'. Staining in embryos was developed at 37 °C until an optimal intensity was observed, embryos were then rinsed briefly in PBS, and then post-fixed in 4% PFA overnight. A minimum of three embryos were analyzed per genotype.

## Real-time PCR

Real-time reverse transcriptase PCR (RT-PCR) was carried out essentially as previously described (*Van Otterloo et al., 2018*). Briefly, embryos were harvested at the indicated stage and facial prominences dissected off for RNA isolation. Tissue was stored in RNAlater at −20 °C until genotyping was completed on samples. Following positive identification of genotypes, tissue was equilibrated at 4 °C for 1 day, RNAlater removed, and RNA extracted from tissue samples using the Rneasy Plus Mini Kit (Qiagen) along with the optional genomic DNA eliminator columns. A similar approach was used for tissue specific RT-PCR analysis. However, once facial prominences were isolated, samples were placed in a 24-well plate with 1 mL of 1 mg/ml Dispase II (in PBS). The samples were incubated with rocking at 37 °C for 30–40 min and then the facial ectoderm carefully dissected away from the mesenchyme into ice-cold PBS, as described (*Li and Williams, 2013*) and RNA extracted from tissue samples using the Rneasy Plus Mini Kit (Qiagen). Following RNA isolation and quantification, cDNA was generated using a set amount of RNA and the SuperScript III First-Strand Synthesis Kit (Invitrogen/ThermoFisher Scientific). Once cDNA was generated, quantitative real-time PCR analysis was conducted using a Bio-Rad CFX Connect instrument, Sybr Select Master Mix (Applied Biosystems, ThermoFisher Scientific) and 20 µl reactions (all reactions performed in triplicate). All primers were designed to target exons flanking (when available) large intronic sequences. Relative mRNA expression levels were quantified using the $^{\Delta\Delta}$Ct method (*Dussault and Pouliot, 2006*) and an internal relative control (e.g. β-actin).

## Scanning electron microscopy

Specimens were processed for electron microscopy according to standardized procedures. Briefly, the samples were fixed in glutaraldehyde, rinsed in sodium cacodylate buffer, and secondarily fixed in osmium tetroxide before dehydrating in a graduated ethanol series. Following dehydration, the samples were mounted on a SEM stub and sputter coated for 30 s using a gold/palladium target in a Lecia (Buffalo Grove, IL) EM ACE 200 Vacuum Coater. Scanning electron micrographs were acquired using a JEOL (Peabody, MA) JSM-6010LA electron microscope operated in high-vacuum mode at 20kV. A minimum of three embryos were analyzed per genotype.

## Acknowledgements

The authors acknowledge Irene Choi, for her care and maintenance of the animal colonies utilized in this study, and Eric Wartchow for assistance with electron microscopy. We are grateful to our colleagues at the University of Colorado, Anschutz Medical Campus, Department of Craniofacial Biology and the University of Iowa, Iowa Institute for Oral Health Research and the Department of Anatomy and Cell Biology for their valuable critiques and feedback. We thank the Genomics Shared Resource at the University of Colorado Cancer Center for assistance with next-generation sequencing. And finally, our funding sources, including the National Institute of Dental and Craniofacial Research (TW, EVO) and the University of Iowa, College of Dentistry and Dental Clinics (EVO).

## Additional information

### Funding

| Funder | Grant reference number | Author |
|---|---|---|
| National Institute of Dental and Craniofacial Research | 2R01 DE12728 | Trevor Williams |

| Funder | Grant reference number | Author |
|---|---|---|
| National Institute of Dental and Craniofacial Research | | Eric Van Otterloo |
| University of Iowa | | Eric Van Otterloo |
| College of Dentistry and Dental Clinics | | Eric Van Otterloo |

The funders had no role in study design, data collection and interpretation, or the decision to submit the work for publication.

### Author contributions

Eric Van Otterloo, Conceptualization, Data curation, Formal analysis, Funding acquisition, Investigation, Methodology, Supervision, Validation, Visualization, Writing – original draft, Writing – review and editing; Isaac Milanda, Data curation, Formal analysis, Investigation; Hamish Pike, Hong Li, Formal analysis, Investigation; Jamie A Thompson, Data curation, Formal analysis, Investigation, Writing – review and editing; Kenneth L Jones, Formal analysis, Resources, Software; Trevor Williams, Conceptualization, Data curation, Formal analysis, Funding acquisition, Investigation, Methodology, Supervision, Writing – original draft, Writing – review and editing

### Author ORCIDs

Eric Van Otterloo  http://orcid.org/0000-0001-5958-5742
Trevor Williams  http://orcid.org/0000-0002-2416-4603

### Ethics

All experiments were conducted in accordance with all applicable guidelines and regulations, following the 'Guide for the Care and Use of Laboratory Animals of the National Institutes of Health'. The animal protocol utilized was approved by the Institutional Animal Care and Use Committee of the University of Colorado - Anschutz Medical Campus (#14) and the Institutional Animal Care and Use Committee of the University of Iowa (#9012197).

### Decision letter and Author response

Decision letter https://doi.org/10.7554/eLife.70511.sa1
Author response https://doi.org/10.7554/eLife.70511.sa2

## Additional files

### Supplementary files

• Supplementary file 1. Summary of gene expression values in the craniofacial surface ectoderm versus the facial mesenchyme of wild-type E11.5 mouse embryos and the association of these genes with the ATAC-seq elements identified in *Figure 2E*, the promoter distal peaks (used for cumulative distribution plotting).

• Supplementary file 2. Summary of ATAC-seq element gene association for the AP-2 dependent peaks. For each gene, the total number of elements (both promoter proximal and distal) and genomic location of each element, relative to the transcriptional start site, are indicated.

• Supplementary file 3. Gene expression summary for E10.5 RNA-seq analysis of control and EDKO facial prominence samples. Note, each tab of the spreadsheet contains a subset of the larger dataset that was used for further analysis.

• Supplementary file 4. Summary of primers used for the current study.

• Transparent reporting form

### Data availability

Sequencing data has been deposited in the Gene Expression Omnibus under accession code GSE199342.

The following dataset was generated:

| Author(s) | Year | Dataset title | Dataset URL | Database and Identifier |
|---|---|---|---|---|
| Van Otterloo E | 2022 | ATAC-seq, histone-seq, and RNA-seq of mouse craniofacial tissues | https://www.ncbi.nlm.nih.gov/geo/query/acc.cgi?acc=GSE199342 | NCBI Gene Expression Omnibus, GSE199342 |

The following previously published datasets were used:

| Author(s) | Year | Dataset title | Dataset URL | Database and Identifier |
|---|---|---|---|---|
| Bing R, ENCODE Consortium | 2017 | ENCODE, heart, ATACseq | https://doi.org/10.17989/ENCSR820ACB | ENCODE, 10.17989/ENCSR820ACB |
| Bing R, ENCODE Consortium | 2017 | ENCODE, liver, ATACseq | https://doi.org/10.17989/ENCSR785NEL | ENCODE, 10.17989/ENCSR785NEL |
| Bing R, ENCODE Consortium | 2017 | ENCODE, hindbrain, ATACseq | https://doi.org/10.17989/ENCSR012YAB | ENCODE, 10.17989/ENCSR012YAB |
| Bing R, ENCODE Consortium | 2017 | ENCODE, midbrain, ATACseq | https://doi.org/10.17989/ENCSR382RUC | ENCODE, 10.17989/ENCSR382RUC |
| Bing R, ENCODE Consortium | 2017 | ENCODE, forebrain, ATACseq | https://doi.org/10.17989/ENCSR273UFV | ENCODE, 10.17989/ENCSR273UFV |
| Bing R, ENCODE Consortium | 2017 | ENCODE, neural tube, ATACseq | https://doi.org/10.17989/ENCSR282YTE | ENCODE, 10.17989/ENCSR282YTE |
| Bing R, ENCODE Consortium | 2017 | ENCODE, kidney, ATACseq | https://doi.org/10.17989/ENCSR023QZX | ENCODE, 10.17989/ENCSR023QZX |
| Bing R, ENCODE Consortium | 2017 | ENCODE, intestine, ATACseq | https://doi.org/10.17989/ENCSR983JWA | ENCODE, 10.17989/ENCSR983JWA |
| Hooper J, Li H, Williams T | 2017 | Facebase RNAseq | https://www.facebase.org/chaise/record/#1/isa:dataset/RID=TJA | Facebase, FB00000867 |
| Li H, Jones KL, Hooper JE, Williams T | 2019 | Single cell RNA-sequencing of E11.5 Crect; ROSA26-lacZ mouse fusing upper lip and primary palate | https://www.facebase.org/chaise/record/#1/isa:dataset/RID=1-5004 | Facebase, FB00001039 |

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
