## [Editor Report]

The TFAP2 transcription factor family is a well known regulator of craniofacial development and evolution. In this study the authors have undertaken a comprehensive analysis of TFAP2a and TFAP2b function in the facial ectoderm. Through genomic analyses, the authors identified key TFAP2 dependent genomic elements in the facial ectoderm, their predicted transcription factor binding profiles. WNT signaling is downregulated in TFAP2mutants and can be rescued by overexpressing Wnt1 in the facial ectoderm. Overall, this study provides new insights into the role of AP2 genes in facial ectoderm signaling and development during craniofacial morphogenesis.

---

## [Decision Letter]

**Decision letter after peer review:**

Thank you for submitting your article "AP-2α and AP-2β cooperatively function in the craniofacial surface ectoderm to regulate chromatin and gene expression dynamics during facial development." for consideration by *eLife*. Your article has been reviewed by 3 peer reviewers, one of whom is a member of our Board of Reviewing Editors, and the evaluation has been overseen by a Reviewing Editor and Kathryn Cheah as the Senior Editor.

Essential revisions:

1) Validation of the genetic mouse models employed is needed. The activity of the ectodermal Cre used in this study (Crect) was shown by Schock et al., 2017 to be variable. It is not clear how variable Crect activity was controlled for in this study and the extent to which this may impact variability. Include images of the Crect recombination pattern, and AP-2a and AP-2b expression at the critical stages analysed in the mutants, as this would facilitate a better understanding of the phenotypes produced and also show whether AP-2a and AP-2b are co-expressed at all times in the facial ectoderm and if Crect encompasses their full expression domains.

2) Can the authors provide an explanation for why the phenotype of the Tfap2b mutant (EBKO) mice is milder than that of the Tfap2a mutant (EAKO) mice or the double knockout (EDKO) mice.

3) Further validate the bioinformatic analysis with in vivo expression analysis of these genes during development. For example, do Tfap2a and Tfap2b expression patterns overlap in the same cells? Are Tfap2a and Tfap2b expressed extensively in the facial prominence epithelium or are they more specifically associated with the most affected regions (e.g. lambdoid junction, mandible fusion site, etc.) in the knockout mice?

4) The authors need to analyze the compensatory mechanism of Tfap2a and Tfap2b during craniofacial development. Is it possible that Tfap2a can compensate for the loss of Tfap2b, while Tfap2b cannot do the same for Tfap2a? The authors need to provide further analysis, such as testing the expression of Tfap2a and Tfap2b in EBKO and EAKO mice, respectively, and compare the data to the control.

5) Co-localize the expression of Tfap2a and Tfap2b with Wnt-related molecules to confirm whether Tfap2 genes in the epithelium regulate Wnt signaling molecules in a cell-autonomous manner or through cell-cell interaction?

6) Axin2-lacZ reporter mice showed decreased expression of Axin2 in both facial ectoderm and mesenchyme in EAKO and EDKO mice. The conclusion that loss of AP-2 proteins in the ectoderm leads to reduced Wnt signaling in both the ectoderm and underlying mesenchyme should be better supported. Confirm reduced Wnt signaling activity in the EDKO using immunofluorescent staining for active β-catenin. To explain whether this decrease is directly caused by decreasing expression of Wnt related molecules in facial ectoderm, the authors should include more data on those genes in the mesenchyme.

7) It's unclear whether Wnt1 overexpressing mice have an endogenous phenotype, and this should have been presented as a control. It's also important to show that overexpressing Wnt1 in the surface ectoderm is sufficient to elicit a Wnt signaling response in the ectoderm and underlying mesenchyme for these mice possibly via Axin2-lacZ, but other downstream markers would be required. Alternatively, conditionally overexpress B-catenin in the facial ectoderm or treat the faces of E11.5 conditional AP-2a and AP-2b mutants in explant culture with exogenous Wnt3a or Wnt9b.

8) The paper would benefit from demonstrating that exogenous Wnt signaling can restore proliferation and patterning, and from additional analyses of potential changes in expression of other signaling pathways in the ectoderm and neural crest cells, known to be associated with syngnathia, maxillo-mandibular patterning and facial clefts.

9) Consider an in vitro cell culture model to test whether Wnt3 and Wnt9b have any role in perturbing the mesenchymal gene expression program. Furthermore, there is a possibility that Wnt1 may not function exactly the same as Wnt3 and Wnt9b. Could the authors confirm that this ectopic Wnt1 can restore the reduced Wnt activity associated with the reduced Wnt3 and Wnt9b in EAKO and EDKO mice?

*Reviewer #1:*

The AP2 transcription factor family is a well known regulator of craniofacial development and evolution. Previous studies revealed the prominent role for AP-2a, and functional redundancy between AP-2a and AP-2b, in neural crest cell and craniofacial development. However questions about the importance of AP-2a and AP-2b signaling and function in the facial ectoderm and their impact on the underlying neural crest cells during craniofacial development. The authors performed a comprehensive analysis of AP-2a and AP-2b in facial ectoderm. By combining wild-type and mutant genomic and transcriptomic assays the authors identified Wnt signaling as a key regulator of AP-2 function. Decreased Wnt signaling was validated via Axin2-lacZ signaling and partial rescue of the phenotype was achieved by overexpressing Wnt1 in the facial ectoderm.

Images of the Crect recombination pattern, and AP-2a and AP-2b expression at the critical stages analysed in the mutants, are needed for context and to facilitate a better understanding of the phenotypes and also show whether AP-2a and AP-2b are co-expressed at all times in the facial ectoderm and if Crect encompasses their full expression domains. Additional phenotypic analyses are required as the mechanism can be explained simply by decreased cell division in the ectoderm or neural crest cells, or via loss of Wnt signaling. It's unclear whether Wnt1 overexpressing mice have an endogenous phenotype, and this should have been presented as a control. It's also important to show that overexpressing Wnt1 in the surface ectoderm is sufficient to elicit a Wnt signaling response in the ectoderm and underlying mesenchyme for these mice possibly via Axin2-lacZ, but other downstream markers would be required. This is because Wnt1 is not endogenously expressed in the facial ectoderm, nor is it equivalent to Wnt3a or Wnt9b. An alternative would be to conditionally overexpress B-catenin in the facial ectoderm or treat the faces of E11.5 conditional AP-2a and AP-2b mutants in explant culture with exogenous Wnt3a or Wnt9b.

There is a mechanistic disconnect between the ectoderm deletion of AP-2a and AP-2b and the molecular changes observed and their cellular effects on the underlying neural crest cell derived mesenchyme. The phenotype cannot be explained simply by decreased cell division (pHH3+) in the ectoderm or neural crest cells, or via loss of Wnt signaling. It's not actually clear those two things are related.

Unfortunately, in the absence of lineage tracing of neural crest cells, it's not clear what happens to those cells, although one might predict based on morphology that neural crest cell formation and migration is normal, but that differentiation is primarily affected. The authors noted the subsequent presence of syngnathia which has been observed in association with altered Foxc1 (Inman et al., 2013), Fgf8 (Inman et al., 2013) signaling in the facial ectoderm and altered Bmp signaling (He et al., 2014) in the underlying neural crest cells. Interestingly the authors have previously shown that modulating Fgf8 gene dosage can ameliorate some of the facial anomalies in AP-2a mice (Green et al., 2015). The authors also observed duplications of Meckel's cartilage which in association with the authors' noted changes in Endothelin signaling may be indicative of maxillary-mandibular patterning defects.

Although it is understandable that the authors took advantage of a Wnt1 overexpressing conditional line to try and restore Wnt signaling, it's unclear whether these mice have an endogenous phenotype, and this should have been presented as a control. It's also important to show that overexpressing Wnt1 in the surface ectoderm is sufficient to elicit a Wnt signaling response in the ectoderm and underlying mesenchyme for these mice via Axin2-lacZ. This is because Wnt1 is not endogenously expressed in the facial ectoderm, nor is it equivalent to Wnt3a or Wnt9b. An alternative would be to conditionally overexpress B-catenin in the facial ectoderm or treat the faces of E11.5 conditional AP-2a and AP-2b mutants in explant culture with exogenous Wnt3a or Wnt9b.

The paper would benefit from demonstrating that exogenous Wnt signaling can restore proliferation and patterning, and from additional analyses of potential changes in expression of other signaling pathways in the ectoderm and neural crest cells, known to be associated with syngathia, maxillo-mandibular patterning and facial clefts.

*Reviewer #2:*

This manuscript, "AP-2α and AP-2β cooperatively function in the craniofacial surface ectoderm to regulate chromatin and gene expression dynamics during facial development," investigated the function of Tfap2a and Tfap2b in facial development, especially focusing on the function of Tfap2a and Tfap2b in ectoderm and how they affect the interplay between surface ectoderm and underlying NCCs. The comprehensive analysis using ATAC-seq and RNAseq in this study provides novel and useful insights for the field; however, more details need to be included for the in vivo analysis. In addition, although Wnt signaling is significantly affected in Tfap2a mutant (EAKO) and Tfab2a/b double knockout (EDKO) mice, those with increasing Wnt1 still have severe craniofacial defects, suggesting that Wnt signaling might not be the sole player in this model.

Specific comments:

1. The phenotype of the Tfap2b mutant (EBKO) mice is milder than that of the Tfap2a mutant (EAKO) mice or the double knockout (EDKO) mice. Can the authors provide an explanation for this result? In addition to showing expression of Tfap2 transcripts it would be very useful to further validate the bioinformatic analysis with in vivo expression analysis of these genes during development. The authors also would need to provide further analysis, such as testing the expression of Tfap2a and Tfap2b in EBKO and EAKO mice, respectively, and compare the data to the control. The authors need to analyze the compensatory mechanism of Tfap2a and Tfap2b during craniofacial development. Since EAKO mice show much more severe defects than EBKO mice, is it possible that Tfap2a can compensate for the loss of Tfap2b, while Tfap2b cannot do the same for Tfap2a?

2. Comprehensive ATAC-seq analysis revealed reduced accessibility of Wnt signaling-related genes. Can the authors co-localize the expression of Tfap2a and Tfap2b with Wnt-related molecules to confirm whether Tfap2 genes in the epithelium regulate Wnt signaling molecules in a cell-autonomous manner or through cell-cell interaction? Does Tfap2b has a different role than Tfap2a, and that only Tfap2a is a key regulator of Wnt signaling?

3. The authors used Wnt1 overexpression mice to rescue the defects in EAKO and EDKO mice, in which Wnt3 and Wnt9b are affected but this does not provide direct evidence for the functional requirement of epithelial-derived Wnt3 and Wnt9b in facial development. there is a possibility that Wnt1 may not function exactly the same as Wnt3 and Wnt9b. Could the authors confirm that this ectopic Wnt1 can restore the reduced Wnt activity associated with the reduced Wnt3 and Wnt9b in EAKO and EDKO mice?

4. Among the expression levels of Tfap2a, Tfap2b, and Tfap2c from E10.5 to E12.5 in previously published data, Tfap2b increased the most from E10.5-12.5. However, the phenotype of the Tfap2b mutant (EBKO) mice is milder than that of the Tfap2a mutant (EAKO) mice or the double knockout (EDKO) mice. Can the authors provide an explanation for this result? In addition to showing expression of Tfap2 transcripts using the previous E10.5-E12.5 RNAseq data of the facial prominence, it would be very useful to further validate the bioinformatic analysis with in vivo expression analysis of these genes during development. For example, do Tfap2a and Tfap2b expression patterns overlap in the same cells? Are Tfap2a and Tfap2b expressed extensively in the facial prominence epithelium or are they more specifically associated with the most affected regions (e.g. lambdoid junction, mandible fusion site, etc.) in the knockout mice?

5. The authors need to analyze the compensatory mechanism of Tfap2a and Tfap2b during craniofacial development. Since EAKO mice show much more severe defects than EBKO mice, is it possible that Tfap2a can compensate for the loss of Tfap2b, while Tfap2b cannot do the same for Tfap2a? The authors need to provide further analysis, such as testing the expression of Tfap2a and Tfap2b in EBKO and EAKO mice, respectively, and compare the data to the control.

6. Comprehensive ATAC-seq analysis was performed in this study, which is useful information for the field. In the ATAC-seq analysis, Wnt signaling-related genes showed reduced accessibility in mutant models. Can the authors co-localize the expression of Tfap2a and Tfap2b with Wnt-related molecules to confirm whether Tfap2 genes in the epithelium regulate Wnt signaling molecules in a cell-autonomous manner or through cell-cell interaction? Furthermore, whole mount in situ hybridization of Wnt3 and Wnt9b nicely showed reduction of those two genes in the facial ectoderm of EAKO and EDKO mice. However, those genes didn't show much reduction in EBKO mutants. Does this suggest Tfap2b has a different role than Tfap2a, and that only Tfap2a is a key regulator of Wnt signaling? Furthermore, EDKO mice showed dramatic changes in genes involved in Wnt signaling. How these two genes synergistically regulate Wnt signaling during craniofacial development deserves more investigation.

7. Axin2-lacZ reporter mice showed decreased expression of Axin2 in both facial ectoderm and mesenchyme in EAKO and EDKO mice. To explain whether this decrease is directly caused by decreasing expression of Wnt related molecules in facial ectoderm, the authors should include more data on those genes in the mesenchyme. Furthermore, the rescue model still shows quite severe defects in the mandibular area. Since the expression levels of Tfap2a and Tfap2b are comparable among the nasal prominences, maxillary prominence and mandibular prominence, and again the expression levels of Axin2 are also similar between the maxillary and mandibular prominences, why are defects in the mandibular area not rescued?

8. The authors used Wnt1 overexpression mice to rescue the defects in EAKO and EDKO mice, in which Wnt3 and Wnt9b are affected. This does not provide direct evidence for the functional requirement of epithelial-derived Wnt3 and Wnt9b in facial development. Could the authors use an in vitro cell culture model to test whether Wnt3 and Wnt9b have any role in perturbing the mesenchymal gene expression program? Furthermore, there is a possibility that Wnt1 may not function exactly the same as Wnt3 and Wnt9b. Could the authors confirm that this ectopic Wnt1 can restore the reduced Wnt activity associated with the reduced Wnt3 and Wnt9b in EAKO and EDKO mice?

*Reviewer #3:*

This study by Van Otterloo et al., investigates the role of AP-2 transcription factors in regulating an ectodermal gene regulatory network that directs formation and patterning of the face. Through genetic knockouts and by identifying genome-wide differences in chromatin accessibility (ATAC-seq) and gene expression (RNA-seq) in EDKO embryos, the authors identify AP-2 proteins as key transcriptional regulators of ectodermal-derived signals, particularly Wnt ligands. They then provide evidence that the facial phenotype in EDKO embryos is in part linked to reduced Wnt signaling by showing that ectoderm-specific expression of Wnt1 partially rescues facial prominence fusion. Overall, the data presented support their conclusion that AP-2-mediated regulation of Wnt ligands in the ectoderm is critical for ectoderm-mesenchyme interactions in facial development. This study will inform how facial ectoderm is mechanistically involved in orofacial clefts and how human mutations TFAP2A and TFAP2B cause syndromes with craniofacial differences. The integration of mouse genetics and genomics in this study is comprehensive; however, there are some aspects of the data analysis that need to be extended and clarified:

1. The conclusion that loss of AP-2 proteins in the ectoderm leads to reduced Wnt signaling in both the ectoderm and underlying mesenchyme should be better supported. The authors state that reduced output of Wnt signaling in the ectoderm was matched by a significant reduction of Axin2-LacZ reporter expression in the underlying mesenchyme. Yet, the whole mount LacZ stains that are used to make this conclusion make it difficult to distinguish ectoderm versus mesenchymal expression. Additionally, Axin2-LacZ is the only means by which Wnt signaling activity is assessed. Since they show Axin2 is itself a target of AP-2 transcription factors, a readout of its expression may be confounding.

This study provides a very comprehensive analysis and I only have a few suggestions to enhance the experimental rigor:

2. Validation of the genetic mouse models employed is needed. The activity of the ectodermal Cre used in this study (Crect) was shown by Schock et al., 2017 to be variable. Using R26R lineage reporter, Schock et al., found that Crect activity is less defined and produces a greatly reduced recombination pattern in 47% of embryos. It is not clear how variable Crect activity was controlled for in this study and the extent to which this may impact variability in (1) Cre recombination efficiency of 4 AP2 alleles (Tfap2a and Tfap2b), and (2) phenotype, chromatin accessibility, gene expression, and Wnt1-mediated rescue. While variability in Cre recombination efficiency is unlikely to change the conclusion that ectodermal AP-2 proteins are necessary for facial development given the severity of the EDKO phenotype, it could certainly impact the interpretation of ATAC-seq (with two biological replicates) and RNA-seq datasets.

3. Confirm reduced Wnt signaling activity in the EDKO using immunofluorescent staining for active β-catenin. Since Axin2 is itself a target of AP-2 transcription factors, using Axin2-LacZ as the only readout of Wnt activity could be problematic.

4. Analyze Axin2-LacZ expression in section. Lines 565-568 state that EDKO embryos show a "more prominent drop in β-gal staining" than EAKO embryos. Lines 569-570 state that β-gal activity from Axin2-LacZ is clearly reduced in mesenchymal population. Both of these points are difficult to appreciate in whole mount.

5. Validate Cre recombination efficiency. Based on Schock et al., 2017, there is a strong concern that Crect-mediated recombination is variable. In addition, Cre-mediated recombination of 4-5 alleles may not be very efficient. This concern could be addressed by (1) comparing the expression of Tfap2a and Tfap2b in control versus EDKO using the RNA-seq datasets already generated and/or immunofluorescent staining, (2) examining expression of the Wnt1-GFP transgene to show its activation in the ectoderm in control and EDKO embryos.

---

## [Author Response]

Essential revisions:1) Validation of the genetic mouse models employed is needed. The activity of the ectodermal Cre used in this study (Crect) was shown by Schock et al., 2017 to be variable. It is not clear how variable Crect activity was controlled for in this study and the extent to which this may impact variability. Include images of the Crect recombination pattern, and AP-2a and AP-2b expression at the critical stages analysed in the mutants, as this would facilitate a better understanding of the phenotypes produced and also show whether AP-2a and AP-2b are co-expressed at all times in the facial ectoderm and if Crect encompasses their full expression domains.3) Further validate the bioinformatic analysis with in vivo expression analysis of these genes during development. For example, do Tfap2a and Tfap2b expression patterns overlap in the same cells? Are Tfap2a and Tfap2b expressed extensively in the facial prominence epithelium or are they more specifically associated with the most affected regions (e.g. lambdoid junction, mandible fusion site, etc.) in the knockout mice?

We have provided additional information in Figure 1, including a new Figure (Figure 1—figure supplement 1) and incorporated pertinent information in the Materials and methods section to address these two issues.

First, regarding the *Crect* transgene, we generated this line and have maintained it in our colony for many years. We quickly noticed that, like many transgenes, variable expression can be evident between generations. Therefore, we choose males that have continued to provide a relatively stable expression pattern in the ectoderm over many generations. Further, we test every new batch of *Crect* males against a Rosa26 reporter line expressing LacZ or GFP to ensure that we only keep males that produce the required expression pattern in numerous embryos. The desired recombination pattern is shown in whole mount in Figure 1—figure supplement 1A, and in sectioned material in Figure 1—figure supplement 1B-D, with reporter expression largely confined to the surface and oral ectoderm, with additional expression noted in otic and optic epithelium. Any animals that produce the mesenchymal pattern reported in Schock et al., (2017) are discarded. Following these procedures, we have observed a consistent set of mutant phenotypes in the EAKO, EBKO and EDKO embryos used in electron microscopy, skeletal staining, and WMISH analysis, as well as in embryos used for DNA and RNA preparation.

We agree with the reviewers that the efficacy and specificity of Cre recombination, as well as the overlap between expression of *Crect*/*Tfap2a*/*Tfap2b* in the ectoderm, are critical issues that require further documentation, and we are now providing additional figures to validate our findings. Data pertinent to the overlap between the expression of *Tfap2a*, *Tfap2b*, and the *Cre* transgene in the ectoderm are shown in both Figure 1 panel B and Figure 1—figure supplement 1, panel E. These panels show feature plots based on the tSNE data from Figure S2 of the paper Li et al., 2019 which used *Crect* to mark the ectoderm with the R26R LacZ reporter. In Figure 1B, we have outlined the various cell types present in the single cell clustering analysis and shown feature plots for *Tfap2a* and *Tfap2b*. The most concentrated expression of the two *Tfap2* genes maps to the surface ectoderm and periderm, but expression is also apparent in mesenchyme as previously reported in Van Otterloo et al., 2018. Blending of these plots illustrates the considerable overlap between expression of these two AP-2 genes in the surface ectoderm and periderm at cellular resolution. Similarly, Figure 1—figure supplement 1 shows how *Cre* is specifically expressed in the ectoderm, and how it significantly overlaps in that tissue layer with *Tfap2a* and *Tfap2b*.

With respect to the efficacy of Crect in targeting *Tfap2a* and *Tfap2b* in the ectoderm we have now included Figure 1—figure supplement 2. The figure shows the results of RNAseq analysis performed on two separate isolated ectoderm samples from control and EDKO mutant embryos examining expression associated with the *Tfap2a* and *Tfap2b* loci. Targeting of the floxed *Tfap2a* allele removes exons 5 and 6, while exon 6 is lost from the floxed *Tfap2b* locus. These data show a >95% reduction in transcripts containing these floxed exons.

We have now amended the manuscript to provide further information concerning *Crect* and the *Tfap2a* and *Tfap2b* genes. Specifically, we have rewritten the first paragraph of the Results section to include a sentence referring to overlap between *Tfap2a* and *Tfap2b* expression in the surface ectoderm “Further mining of single cell RNA-seq data derived from facial prominences indicated that *Tfap2a* and *Tfap2b* expression also displayed significant overlap within cells of the surface ectoderm and periderm” and changed Figure 1 to show these data.

We have also added a new section in the Materials and methods section concerning the mouse strains:

“Although the Crect transgene is used here to target the early embryonic ectoderm, a previous report has indicated that it frequently produces a broader pattern of recombination (Schock et al., 2017). We used several approaches to avoid this broader expression pattern. First, the *Crect* transgene was always introduced into the experimental embryos via the sire to reduce global recombination sometimes seen with transmission from the female. Second, all sires were tested using reporter lines such as (Gt(ROSA)26Sor^tm1Sor^) or mT/mG, Gt(ROSA)26Sor^tm4(ACTB-tdTomato,-EGFP)Luo/J^ to ensure that they consistently produced the desired pattern of recombination before they were used to generate EAKO, EBKO, or EDKO animals (Figure 1—figure supplement 1). We also confirmed the overlap between the expression of *Cre*, *Tfap2a*, and *Tfap2b* in the ectoderm by mining a previously published single cell RNAseq dataset (Li et al., 2019). Lastly, we determined that the *Crect* transgene was highly efficient at targeting the *Tfap2a* and *Tfap2b* loci based upon RNA expression associated with these genes in Cre expressing cells (Figure 1—figure supplement 2).”

2) Can the authors provide an explanation for why the phenotype of the Tfap2b mutant (EBKO) mice is milder than that of the Tfap2a mutant (EAKO) mice or the double knockout (EDKO) mice.4) The authors need to analyze the compensatory mechanism of Tfap2a and Tfap2b during craniofacial development. Is it possible that Tfap2a can compensate for the loss of Tfap2b, while Tfap2b cannot do the same for Tfap2a? The authors need to provide further analysis, such as testing the expression of Tfap2a and Tfap2b in EBKO and EAKO mice, respectively, and compare the data to the control.

We have considered several scenarios that could account for these observations, but a thorough experimental understanding of this phenomenon is beyond the scope of the current manuscript. One possibility is that there might be subtle differences in the temporal or spatial expression profiles of these factors. An alternative possibility is that there are differences in the mRNA and/or amino acid sequence that might differentially affect stability or localization of the transcripts or the proteins. Third, even though the proteins are quite similar in overall sequence and binding specificity, they may nevertheless interact with suites of unique and/or overlapping co-factors with AP-2α having more critical interactions than AP-2ß. A further possibility is that there may be cross-regulation between *Tfap2a* and *Tfap2b* at some level. To examine this last possibility, as requested, we have performed preliminary studies to examine the expression of *Tfap2a* and *Tfap2b* in the ectoderm of both EAKO and EBKO mice. However, these studies have not produced a definitive answer concerning compensation caused by loss of one or other transcription factor. Therefore, the mechanism(s) by which AP-2α has the more critical function remains unclear. We believe it will require considerable time and effort to perform studies that will definitively test any of these hypothesis—for example by substituting *Tfap2a* into the *Tfap2b* locus and vice versa—and that this is beyond the scope of the current analysis. At the same time, we also note that there are multiple examples throughout the literature where TFAP2 paralogs function redundantly/cooperatively, yet loss of *Tfap2a* alone provides a more dramatic phenotype than deletion of an alternative paralog. For example, we found that, while AP-2α and AP-2ß function cooperatively in the neural crest, loss of AP-2α alone in the neural crest results in a more severe phenotype than loss of AP-2ß alone with respect to both craniofacial development and melanocyte biology (Van Otterloo et al., 2018 and Seberg et al., 2017). This phenomenon has also been noted in the zebrafish model system (e.g., *tfap2a*/*tfap2c*, Li and Cornell, 2007; *tfap2a*/*tfap2e*, Van Otterloo et al., 2010).

However, we have now added the following short statement in the discussion to provide possible explanations for why the phenotype of the *Tfap2b* mutant (EBKO) mice is milder than the *Tfap2a* mutant (EAKO) mice.

“The reasons behind the more prominent function of AP-2a in these systems remains unclear, especially since the two proteins are over 70% identical and bind to the same DNA recognition sequences. Nevertheless, it is possible that some alterations in the mRNA and/or amino acid sequence could differentially affect stability, localization, or interaction with cofactors. Alternatively, there might be subtle differences in the timing, distribution, or levels of functional AP-2α and AP-2ß protein in these tissues that account for the changes in susceptibility to pathology. Further studies will be needed to determine how the consequence of Tfap2a loss is greater than Tfap2b.”

5) Co-localize the expression of Tfap2a and Tfap2b with Wnt-related molecules to confirm whether Tfap2 genes in the epithelium regulate Wnt signaling molecules in a cell-autonomous manner or through cell-cell interaction?6) Axin2-lacZ reporter mice showed decreased expression of Axin2 in both facial ectoderm and mesenchyme in EAKO and EDKO mice. The conclusion that loss of AP-2 proteins in the ectoderm leads to reduced Wnt signaling in both the ectoderm and underlying mesenchyme should be better supported. Confirm reduced Wnt signaling activity in the EDKO using immunofluorescent staining for active β-catenin. To explain whether this decrease is directly caused by decreasing expression of Wnt related molecules in facial ectoderm, the authors should include more data on those genes in the mesenchyme.

We have added two new Supplementary figures to address (1) the issue of the overlap between the AP-2 transcription factors and WNT signaling pathway genes in the ectoderm and (2) how loss of AP-2 in the ectoderm affects the expression of WNT pathway genes in the mesenchyme.

The first figure (Figure 4—figure supplement 1) shows feature plots derived from the E11.5 single cell sequencing data (Li et al., 2019) for multiple *Wnt* ligands and associated genes involved in WNT pathway regulation. In addition, we show the overlap with *Tfap2a* expression for each of these 18 genes in the whole dataset and the surface ectoderm. (*Tfap2b* expression is not included due to the considerable overlap between *Tfap2a* and *Tfap2b* shown in new Figure 1B). Overall, the data show that the ligands *Wnt3*, *Wnt4*, *Wnt6*, *Wnt9b* and *Wnt10a*, as well as other WNT pathway modulators including *Kremen2* and to a lesser extent *Dkk4* are almost exclusively expressed in the ectoderm and that the majority of cells expressing these ligands also co-express the *Tfap2* genes in the surface ectoderm. The data on ectodermal bias in expression for these genes are in agreement with the analyses of Hooper et al., 2020 which used RNAseq analysis to examine expression in the facial ectoderm and mesenchyme compartments. We used the analysis of Hooper et al., as shown in Supplementary File 3 to stratify genes according to ectodermal/mesenchymal ratios of expression in Figure 7E and Figure 7—figure supplement 1.

To summarize, our WMISH, RT-PCR and RNAseq findings demonstrate that there is a significant reduction in the ectodermal expression of *Wnt* related molecules in the EDKO mutants compared to controls (Figures 4, 7, Figure 4—figure supplement 4 and Figure 7—figure supplement 2). Further, there is significant overlap for many of these genes with *Tfap2* expression in the ectoderm and we also we show that several of them are associated with AP-2 dependent ATAC-seq peaks (Figures 3 and 4). Together, these findings would support a hypothesis that regulation of these genes is direct and cell autonomous in nature. However, definitive proof that regulation is exclusively cell autonomous will require considerable further experimentation.

The second figure (Figure 7—figure supplement 3) provides further information regarding how *Wnt* pathway gene expression is affected in the mesenchyme when *Tfap2a* and *Tfap2b* are lost from the ectoderm. In addition to the information concerning the reduction of *Axin2* expression which we reported in Figure 4 and Figure 4—figure supplement 4, we now show results from RT-PCR analysis of 5 other *Wnt* pathway genes that show significant expression in the mesenchyme. Samples of facial prominence mesenchyme were collected from E11.5 control and mutant embryos, RNA extracted, cDNA synthesized, and real-time PCR conducted on these 5 transcripts. We identified an ~ 2 to 5 fold reduction in expression of all 5 transcripts within the mesenchyme of EDKO’s versus controls. These findings are consistent with loss of ectodermal *Tfap2* having a non-cell autonomous impact on these *Wnt*-associated genes in the mesenchyme.

We have amended the text of the results in several places to incorporate the data included in these two new Supplemental figures.

7) It's unclear whether Wnt1 overexpressing mice have an endogenous phenotype, and this should have been presented as a control. It's also important to show that overexpressing Wnt1 in the surface ectoderm is sufficient to elicit a Wnt signaling response in the ectoderm and underlying mesenchyme for these mice possibly via Axin2-lacZ, but other downstream markers would be required. Alternatively, conditionally overexpress B-catenin in the facial ectoderm or treat the faces of E11.5 conditional AP-2a and AP-2b mutants in explant culture with exogenous Wnt3a or Wnt9b.

We have added a new Supplementary figure (Figure 8—figure supplement 1) to address the issue of the endogenous *Crect Wnt1* overexpression (OX) phenotype. These images show the phenotype of E12.5 control and *Wnt1* overexpression mice, a time point when facial fusion has occurred in both genotypes. Although we have not analyzed the *Wnt1* overexpression phenotype in detail, there are certain noticeable differences from controls. First, the angle between the forebrain and snout is more pronounced in the overexpressing animals. Second, the eye is abnormal in overexpressing mice, with an apparent suppression of lens formation. Note that at later time points the *Crect Wnt1* OX animals develop severe overall edema and do not survive until term. We have not explored these phenotypes further as it is not the focus of the manuscript.

With respect to the other suggestions, we have also examined the *Crect Wnt1* OX phenotype in combination with *Axin2-LacZ*, but do not see major quantitative or qualitative differences in staining. We also note that there were no appreciative changes in b-catenin protein distribution between mutants and controls. These observations fit with our current working hypothesis for many types of cleft lip and primary palate—that just subtle changes in signaling can prevent normal attachment of the prominences at the lambdoid junction. As a result, subsequent continued growth of the prominences—unrestrained by the absence of fusion—causes major morphological defects. Lastly, in respect to this model, the eye phenotype resembles that observed in the study by Smith et al., 2005 (Dev Biol, vol 285, pp 477-489, “The duality of β-catenin function: a requirement in lens morphogenesis and signaling suppression of lens fate in periocular ectoderm”). In this study, a form of ß-catenin that is unable to be degraded is expressed in the presumptive lens placode and this gain of function in WNT signaling also suppresses lens formation.

We also considered using expression of stabilized ß-catenin under the control of the *Crect* transgene to rescue the phenotype. In this respect, we have previously conditionally overexpressed the stabilized form of ß-catenin in the surface ectoderm using *Crect* (Dev Biol (2011), vol 349 pp 261-269). However, facial and vascular defects are more severe in this ß-catenin model, with the mice failing to survive beyond mid embryogenesis and so they are not as useful for the current analysis. We also note that stabilization of ß-catenin in the ectoderm is not equivalent to overexpression of the *Wnt1* ligand; the former will have its primary effect on the ectoderm, whereas the latter will act both on the ectoderm in a paracrine fashion, but also on the underlying mesenchyme. Finally, we appreciate the suggestion of using explant culture to study the interaction between the AP-2 transcription factors and the WNT pathway, and we hope to explore this in the future, but we believe that this is outside the scope of the current manuscript, especially since we postulate that changes in the overall morphogenesis of the head are involved in the AP-2 dependent pathology.

In addition to the new Supplemental Figure, we have added the following statement in the Results section:

“First, though, we examined how Crect mediated Wnt1 overexpression in the ectoderm might impact face development to assess its suitability as a rescue model (Figure 8—figure supplement 1). In common with controls, E12.5 Crect Wnt1^ox^ embryos had completed fusion of the face to form an intact upper lip. However, there were developmental changes in that mutant animals had a more pronounced angle between the forebrain and snout than controls and there were also defects in eye formation, consistent with activation of the Wnt pathway in this process (Smith et al., (2005)). Nevertheless, based on the overall facial phenotype, we reasoned that this approach was feasible to supplement Wnt ligand expression in the facial ectoderm of the EAKO and EDKO mice.”

8) The paper would benefit from demonstrating that exogenous Wnt signaling can restore proliferation and patterning, and from additional analyses of potential changes in expression of other signaling pathways in the ectoderm and neural crest cells, known to be associated with syngathia, maxillo-mandibular patterning and facial clefts.

The analyses we have performed to date point to the WNT pathway as being the most severely affected signaling system that is both heavily altered in our model, associated with craniofacial clefting defects, and able to rescue aspects of the mutant phenotype. Therefore, we have highlighted this pathway in our studies. Nevertheless, we cannot exclude other mechanisms contributing to the phenotype—either alone, or in association with the WNT pathway.

At the level of RNA transcript analysis, we report in Table 1 changes in components of CXCL, EDN, and FGF signaling, but these do not appear to occur at levels that would account for the observed phenotypes (this is mentioned within the Results section). With respect to jaw patterning and syngnathia, we do not find any major differences at E10.5 in the expression levels of genes previously associated with causing similar defects in these tissues such as *Foxc1*, *Six1*, *Bmp4*, *Fgf8* or *Dlx* family members. As noted, there was some slight change in *Ednra* expression, and a major change in the level of *Gbx2* (which has previously been reported to be a target of DLX5/6), but at this point nothing that would account for the observed jaw phenotypes. The analysis of the roles of AP-2 in jaw patterning and syngnathia are the subject of ongoing studies in our laboratories and our current data indicate that this likely utilizes a mechanism parallel to the DLX, EDN, SIX, and FOXC1 pathways.

9) Consider an in vitro cell culture model to test whether Wnt3 and Wnt9b have any role in perturbing the mesenchymal gene expression program. Furthermore, there is a possibility that Wnt1 may not function exactly the same as Wnt3 and Wnt9b. Could the authors confirm that this ectopic Wnt1 can restore the reduced Wnt activity associated with the reduced Wnt3 and Wnt9b in EAKO and EDKO mice?

We appreciate the reviewers’ suggestion to consider an in vitro model in examining WNT3 and WNT9B in more detail and agree that these specific WNTs are good candidates given their previously reported involvement in craniofacial development (Carroll et al., 2005; Juriloff et al., 2006; Niemann et al., 2004; Chiquet BT et al., 2008; Yao T et al., 2011; Fontoura C et al., 2015). However, we note that the expression of a wide range of *Wnt* ligands are impacted in our model, suggesting the likelihood of a combinatorial influence of multiple WNTs on the phenotype observed. Further, given WNT3/WNT9B’s activation of the canonical WNT pathway (Carroll et al., 2005) and WNT1’s primary ability to activate this pathway, including previous evidence of WNT1’s ability to substitute for WNT9B signaling (Carroll et al., 2005), the utilization of over-expressed WNT1 we feel is well justified. We further note that the Wnt1 OX model has also been used by Ferretti et al., (Dev Cell, vol 621, pp 627-641 (2011) *“*A Conserved Pbx-Wnt-p63-Irf6 Regulatory Module Controls Face Morphogenesis by Promoting Epithelial Apoptosis*”*) as a Wnt rescue experiment for a Pbx-based CL/P model.